# Airborne DNA reveals predictable spatial and seasonal dynamics of fungi

Fungi are among the most diverse and ecologically important kingdoms in life. However, the distributional ranges of fungi remain largely unknown as do the ecological mechanisms that shape their distributions[1,2]. To provide an integrated view of the spatial and seasonal dynamics of fungi, we implemented a globally distributed standardized aerial sampling of fungal spores[3]. The vast majority of operational taxonomic units were detected within only one climatic zone, and the spatiotemporal patterns of species richness and community composition were mostly explained by annual mean air temperature. Tropical regions hosted the highest fungal diversity except for lichenized, ericoid mycorrhizal and ectomycorrhizal fungi, which reached their peak diversity in temperate regions. The sensitivity in climatic responses was associated with phylogenetic relatedness, suggesting that large-scale distributions of some fungal groups are partially constrained by their ancestral niche. There was a strong phylogenetic signal in seasonal sensitivity, suggesting that some groups of fungi have retained their ancestral trait of sporulating for only a short period. Overall, our results show that the hyperdiverse kingdom of fungi follows globally highly predictable spatial and temporal dynamics, with seasonality in both species richness and community composition increasing with latitude. Our study reports patterns resembling those described for other major groups of organisms, thus making a major contribution to the long-standing debate on whether organisms with a microbial lifestyle follow the global biodiversity paradigms known for macroorganisms[4,5].

Global biodiversity of microorganisms and the factors determining their distribution and activity remain poorly known despite their major ecological and economic importance in various ecosystems[6–8]. Recently developed technologies and analytical methods provide groundbreaking opportunities for both the improved sampling of biodiversity and unravelling how biodiversity is structured at large spatial and temporal scales[9–11]. These new methods thus provide the opportunity to uncover previously unmapped biodiversity patterns of microbial communities and to discover the ecological processes that shape their diversity at the global scale.

Fungi are among the most diverse and ecologically important living organisms. They mediate crucial processes in terrestrial ecosystems as decomposers of dead tissues (saprotrophs), mutualistic partners (ectomycorrhizal, ericoid, endophytic and lichenized fungi) and as pathogens of almost all terrestrial multicellular organisms. In spite of its importance, fungal diversity remains poorly explored[1]. Although roughly 156,000 species of fungi have been scientifically described and recognized as valid to date[12], estimates of global species richness vary from 0.5 to 10 million[13,14]. Consequently the global spatial and temporal distributions of fungi remain largely unknown. Recently developed DNA-based survey methods have greatly improved our knowledge of large-scale patterns of fungal diversity[15–19]. Soil sampling has proved particularly popular, driven by an interest in the key functions of soil fungi as plant symbionts and nutrient cyclers[2,16,18,20]. Nevertheless it remains to be seen whether patterns in soil-borne fungi reflect patterns in other fungal taxa, or indeed in general biodiversity[21]. In fact, studies targeting different fungal groups have produced disparate results. Tedersoo et al.[16] found that, although overall fungal diversity in soil increases toward the Equator, this pattern does not apply to ectomycorrhizal fungi, which are most diverse in boreal and temperate regions. However, a meta-analysis of metabarcoding data from soil- and root-associated fungi reported that total fungal diversity is higher at higher latitudes[19]. Among further disparities, the diversity of leaf-associated aquatic fungi has been found to peak at mid-latitudes[22] whereas that of terrestrial leaf endophytes increases in tropical regions[23].

Local studies conducted in both Arctic and temperate environments have shown that fungal activity presents pronounced seasonal variation[24–28] whereas a study conducted in the tropics showed no such variation[29], suggesting that seasonality may be latitude dependent. However, most large-scale surveys of fungi have included limited temporal replication of the same locations, leaving a major knowledge gap about their global seasonal dynamics. The few larger-scale studies that involve temporal replication include meta-analyses on heterogeneous datasets[30,31] or historic records of fruiting-body occurrences[32]. The general conclusion drawn from these studies is that the composition and biomass of fungal communities follow the phenology of their hosts and seasonal changes in precipitation and temperature. Hence, the lack of controlling for effects of local seasonal variation may have also confounded some conclusions on the global spatial patterning of fungal diversity.

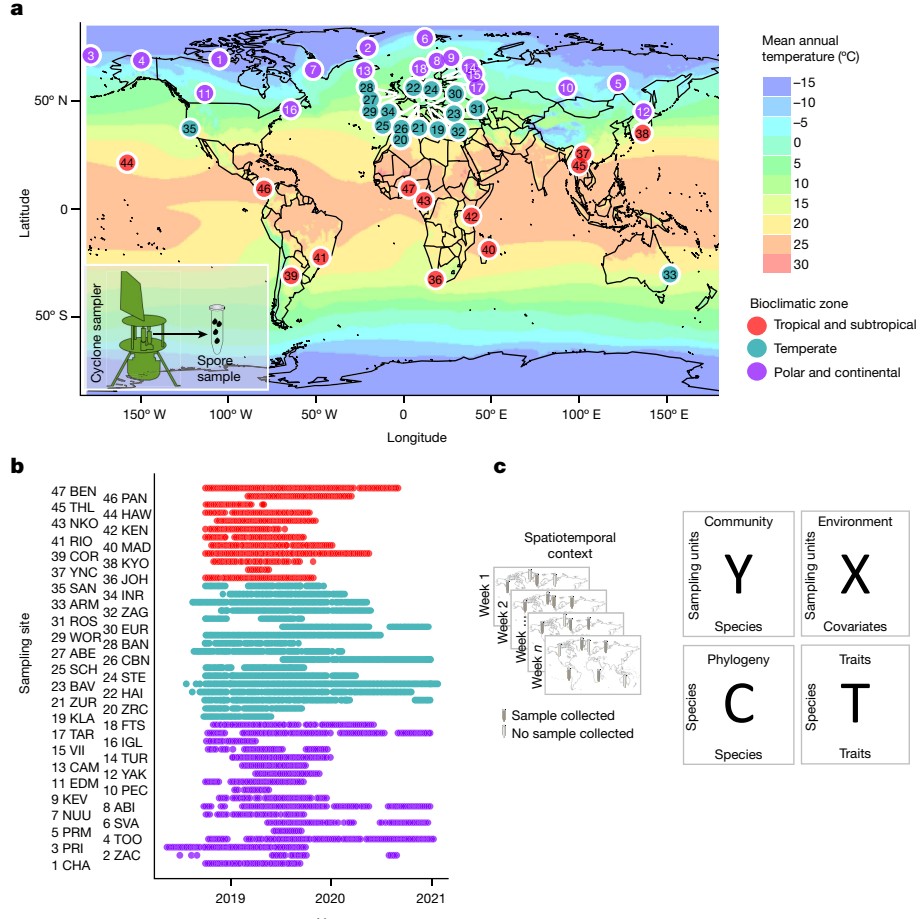

**Fig. 1 | GSSP study design and data. a**, The sampling sites include locations in the tropical-subtropical (red), temperate (cyan) and polar-continental (purple) climatic zones, shown here superimposed on a map of MAT. Airborne fungal samples were collected by a cyclone sampler, each sample consisting of fungal spores filtered from 24 m[3] of air during the 24 h sampling period. **b**, The study design included weekly samples taken over 1–2 years, with some variation among sites due to logistical constraints. The site name abbreviations (three-letter codes next to the site numbers) correspond to those used in the published data[59]. **c**, The data-generation pipeline produced data matrices that were used for the ecological analyses: the spatial and temporal coordinates of the samples, species occurrence data (Y), climatic and weather data (X), fungal guild and spore size data (T) and taxonomic affiliations serve as a proxy for phylogenetic relationships (C).

A recent methodological breakthrough regarding the surveying of fungi consists of sampling fungal spores (and other airborne particles, which may include fungal structures such as hyphae and soredia) from the air, followed by DNA sequencing and sequence-based species identification[33]. Air sampling has shown higher diversity and stronger ecological signals in community composition than soil sampling[34]. The feasibility of air sampling to investigate global patterns of fungal diversity was recently demonstrated[35]. Because this method captures airborne fungal spores, it depicts reproduction and dispersal at high temporal resolution. Here we report on the application of air sampling for fungal spores in a new initiative called the Global Spore Sampling Project (GSSP)[3]. The GSSP involves 47 sampling locations distributed across all continents except Antarctica, each location collecting two 24 h samples per week over 1 year or more (Fig. 1a,b). Although the European temperate region is overrepresented in the data, the sampling locations also include Arctic, temperate and tropical areas from other regions (Fig. 1a). As described in detail in ref. 3, we targeted DNA sequencing to a part of the nuclear ribosomal internal transcribed spacer (ITS) region, which is the universal molecular barcode for fungi[36]. However, we note that for some fungal taxa other markers are better suited, such as the nuclear small subunit ribosomal RNA gene fragment for arbuscular mycorrhiza[37]. We applied a DNA spike-in to generate quantitative estimates of change in the amount of DNA[35].

To convert sequence data to species data we denoised the former to form amplicon sequence variants (ASVs)[38], applied probabilistic taxonomic placement using Protax[39,40] and used constrained dynamic clustering to group these ASVs into species-level operational taxonomic units (OTUs)[41]. These OTUs were then classified into previously known versus unknown taxa at all taxonomic levels from phylum to species[3]. To link spatiotemporal patterns in species composition to the ecological drivers behind them, we complement here the fungal species data derived from DNA analyses with environmental and trait data (Fig. 1c). Trait data were compiled using guild and spore size data from several sources (Methods), and environmental data include time- and site-specific climatic data from the Copernicus Climate Change Service Climate Data Store[42].

The fully standardized sampling of fungi at unprecedented spatial and temporal scales enabled an integrated analysis of the ecological drivers behind the spatial and seasonal patterns of global fungal diversity. To achieve this, we first examined how fungal communities differ among the major bioclimatic zones and the extent to which climatic variables explain such differences. We expected to find a clear differentiation in community composition among the main bioclimatic zones, although we expected the spatial differentiation of airborne spores to be less pronounced than previously reported in soil-based studies[16,19] because microscopic propagules can be expected to mix

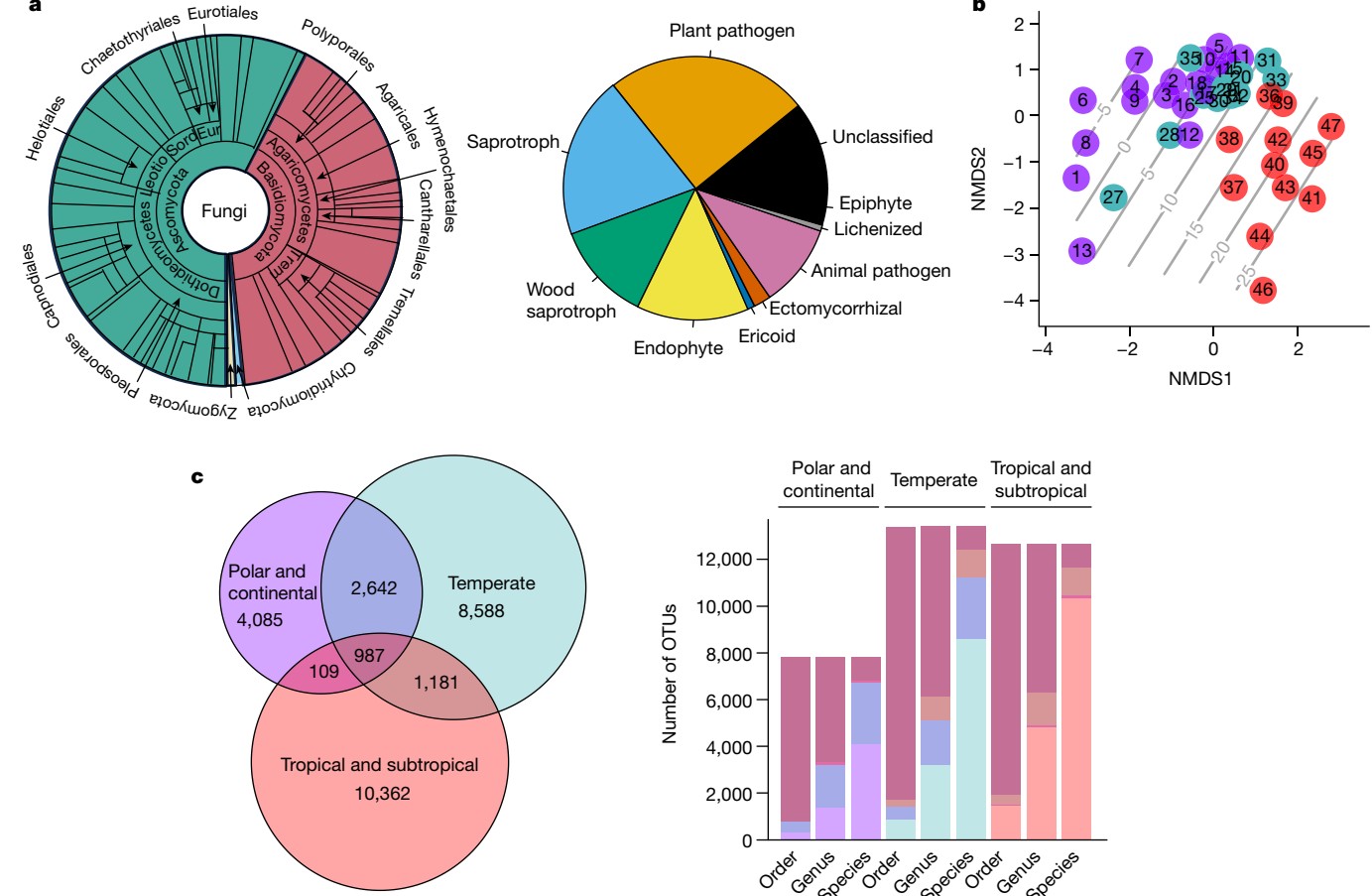

**Fig. 2 | Taxonomic, functional and spatial variation in airborne fungal diversity. a**, Taxonomic and functional guild composition of the data as weighted by prevalence (that is, the number of samples from which the taxon was found). Taxonomic composition is shown for the levels of phylum, class and order. Trophic guild composition is shown based on ref. 54. **b**, Variation in the composition of the fungal community among sites illustrated in the NMDS ordination space, with contour lines representing the MAT (°C) of the site.

**c**, Venn diagram showing the number of OTUs that were distinct or shared among the three major climatic zones included in our study. Note that shown are raw numbers that do not control for the somewhat smaller sampling effort in the tropical-subtropical zone (Fig. 1b). The bar chart shows the number of OTUs that belonged to a genus or order that was either distinct or shared among the three climatic zones. Note that the species-level bars replicate the patterns shown in the Venn diagram.

more readily in air (although samples were collected close to the ground, and often within habitats with limited air flow compared with open areas). Second, we examined how global seasonal patterns of airborne fungi vary with latitude and weather conditions. We expected higher levels of seasonality in species richness and amount of fungal DNA towards higher latitudes, where resources are available for shorter periods of time and where local weather conditions may have a stronger effect on reproductive phenology[32]. Finally we examined whether the ecological drivers shaping the composition of fungal communities translate into predictable variation in species-level traits. To this end we asked whether species' responses to climatic and seasonal factors are phylogenetically and functionally structured. As relevant traits we considered fungal guild[16,43] and spore size[44,45]. We expected to find higher seasonality in host-dependent guilds (pathogenic and symbiotic fungi) than in free-living guilds (saprotrophs), but that spatial patterns of seasonality should be consistent across guilds. We expected to find predictable seasonal variation in spore size, reflecting taxonomic turnover throughout the seasons. Finally, because earlier research has found phylogenetic niche conservatism reflected in the large-scale biogeography of soil fungi[46], we expected to find a phylogenetic signal on the responses of air-fungal communities to the environmental factors that influence their large-scale distributions.

## Climatic effects on spatial distribution

Our samples of airborne fungi include all major taxonomic groups (Fig. 2a). However, some fungal groups are overrepresented and others underrepresented as compared with previously reported patterns among soil fungi. The air samples are particularly rich in plant pathogens, general saprotrophs and wood saprotrophs whereas other common groups such as ectomycorrhizal and lichenized fungi are relatively poorly represented (Fig. 2a).

Among the 27,954 species-level OTUs detected in this study, only 3.5% were observed in all three climatic zones (Fig. 2c). As expected, sampling locations in the polar-continental zone shared the fewest species with sampling locations in the tropical-subtropical zone. However, most order-level taxa were present in all three climatic zones (Fig. 2c). Such an increase in taxonomic overlap among regions with increasing taxonomic rank is also reflected by the stability of the proportions of species belonging to different phyla, with the proportion of Ascomycota spp. being 55–59% and that of Basidiomycota spp. being 38–43% within each of the three climatic zones.

Among the ten most prevalent genera in our data (Extended Data Table 1), seven belonged to the phylum Ascomycota (out of which four belonged to the order Pleosporales) and three to Basidiomycota (out of which two to the order Tremellales). Overall, the three most

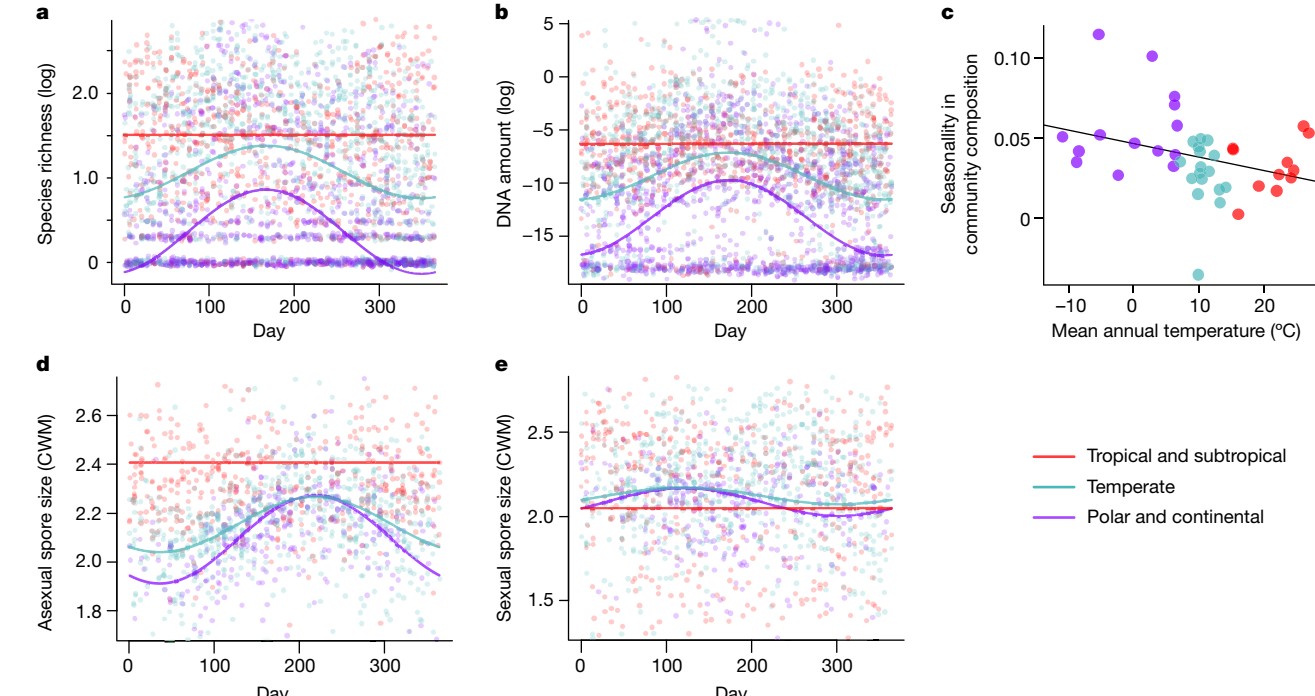

**Fig. 3 | Seasonal variation in airborne fungal diversity. a–e**, The lines representing species richness (**a**), DNA amount (**b**), community composition (**c**) and CWM of asexual (**d**) and sexual (**e**) spore size show the predictions of the best-supported linear mixed models (Methods) for tropical-subtropical (red), temperate (cyan) and polar-continental (purple) climatic zones. Note that the predictions are shown for the Northern Hemisphere whereas for the Southern Hemisphere the seasonal patterns would be mirror images. For community composition (**c**), seasonality for each site is defined as the difference in Jaccard index between samples taken in the same season versus those taken in different seasons (Methods). **a,b,d,e**, The dots representing the raw data have been slightly jittered to show overlap. The line in **c** shows that seasonality in community composition was higher at colder sites (linear regression, $P = 0.04$).

prevalent genera were the ascomycetes *Cladosporium*, *Ascochyta* and *Alternaria*. Genera included in the list of the ten most prevalent genera in all three climatic zones were the ascomycetes *Cladosporium*, *Ascochyta, Alternaria* and *Aureobasidium* and the basidiomycete *Cryptococcus*.

Species composition of local fungal communities was most strongly affected by the mean annual air temperature (MAT) of the site, which, when used as the sole environmental predictor, explained 78% of the deviance in ordination space (Fig. 2b and Extended Data Fig. 1). By comparison, mean annual precipitation (MAP) at the site explained 42% and the mean aridity index (MAI) 35%, whereas mean annual wind speed—which could have added to the mixing of spores to the atmosphere—did not explain much of the deviance (22%). We then compared the relative importance of differences in MAT (selecting for species with similar environmental preferences) and differences in space (probably reflecting the potential for dispersal between two sites, as well as other environmental conditions not considered in the analyses). Because spatial and environmental distances were correlated, we disentangled the effects of these by partitioning variance in community dissimilarity. We found the direct contribution of spatial distance to be 12%, that of climatic distance (derived from MAT) to be 7% and their shared contribution to be 22%. When repeating the analyses with climatic distances derived from MAP (or MAI), the direct contribution of spatial distance was 29% (27%), that of climatic distance 2% (0%) and their shared contribution to be 6% (7%). Hence MAT, rather than MAP or MAI, turned out to be a key driver in determining the large-scale distributions of airborne fungi.

## Seasonal patterns and weather responses

Within airborne spore communities, both OTU diversity and DNA amount increased towards the Equator (Fig. 3a,b). This result was robust with respect to seasonality, because tropical-subtropical sites hosted a greater diversity of fungal species and greater amounts of DNA than temperate and polar-continental sites at all times of the year (Fig. 3a,b). In terms of temporal patterns, seasonal variation in both DNA amount and species richness increased as expected with distance from the Equator, being highest in the Arctic (Fig. 3a,b). During winter at the polar-continental sites, few air samples had detectable levels of fungal DNA and the amount of DNA and number of species both showed a sharp peak during the growing season (Fig. 3a,b). In samples from temperate sites, fungal DNA was found throughout the year but its amount increased markedly from spring to autumn, with the lowest values in winter. In tropical-subtropical sites, fungal DNA amount was high throughout the year. The composition of the fungal community followed the same pattern: in polar-continental sites there was greater turnover in species composition from spring to autumn than in tropical regions during a comparable period (Fig. 3c). However, a comparison of linear mixed models fitted to the data on DNA amount and species richness (Supplementary Information) showed that, although the effect of seasonality generally increased with latitude, the exact timing and amplitude of seasonal variation also had a site-specific component. Thus, although we found that the phenology of fungal spore production is largely consistent within each latitudinal zone, the site-specific component suggests that local factors also play a role in controlling the timing of sporulation. Regarding the effects of weather, we found that both the amount of DNA and observed species richness were generally higher for warm and windy sampling days (Supplementary Information). Whereas most trophic guilds followed the same pattern as overall species richness, endophytes and lichenized species showed higher richness on days with little precipitation. These results were consistent across all latitudes in the sense that, for all but one response variable, the best-supported model was that of constant weather effects (model W1; Methods).

## Phylogenetic and functional structure

The proportion of fungal occurrences for which we had at least family-level information about asexual (respectively, sexual) spore volume varied between 72 and 74% (respectively, 68–70%) among the three climatic zones. However, species-level information was more frequent in the polar-continental and temperate zones (7–8% for asexual and 12–13% for sexual spores) than in the tropical zone (8% for asexual and 5% for sexual spores). Assuming that the detected species were in the asexual stage, these were largest in the tropical-subtropical zone whereas, assuming that the spores were in the sexual stage, these were largest in the temperate and polar-continental zones (Fig. 3d,e). In temperate and polar-continental zones, spore sizes showed marked seasonality, the mean asexual spore size peaking in the autumn and the mean sexual spore size in spring (Fig. 3d,e). This difference between asexual and sexual spores prevailed across all species and within Basidiomycota, but not within Ascomycota (Supplementary Information and Extended Data Fig. 2).

Following the main patterns found for total fungal species richness, all fungal guilds exhibited strong seasonality in species richness in the polar-continental and temperate zones (Supplementary Information and Extended Data Fig. 3). Most guilds were more abundant in the tropics even during the peak season, with the exceptions of ericoid mycorrhizal, ectomycorrhizal and lichenized fungi, which were most abundant in the temperate region (Extended Data Fig. 3).

To determine how the phylogenetic relatedness of fungal species affects global distribution and sporulation patterns, we performed Hierarchical Modelling of Species Communities (HMSC) analysis[47] in which we used as a proxy for the phylogenetic tree a taxonomy of OTUs at the levels of kingdom, phylum, class, order, family, genus and species[3]. Even if this model included only MAT and seasonality as predictors, it reached a high explanatory power (averaged over species, mean area under the curve = 0.90, mean Tjur's $R^2$ = 0.16). This analysis showed variation in the strength of phylogenetic signal among how species responded to focal environmental predictors. Among the species-level responses to environmental conditions, climatic sensitivity showed a moderate phylogenetic signal (Pagel's $\lambda$ = 0.28, $P$ = 4 × 10$^{-12}$), as illustrated by groups of highly related species that showed high or low climatic sensitivity (red and blue bands, respectively, in Fig. 4a in the climatic sensitivity column)−for example, the orders Agaricales and Helotiales being little influenced by climate (Fig. 4b). By contrast, the optimal MAT of the site at which the probability of species occurrence is predicted to be maximized did not show any phylogenetic signal (Pagel's $\lambda$ = −0.01, $P$ = 0.81). Thus some species within the same group preferred colder temperatures whereas others preferred warmer temperatures (Fig. 4a). When we measured the seasonal sensitivity of species by the proportion of variation in species occurrence explained by latitude-dependent seasonality, we observed a strong phylogenetic signal (Pagel's $\lambda$ = 0.39, $P$ = 2 × 10$^{-16}$). In particular, species within the orders Polyporales and Erysiphales showed pronounced seasonal dynamics whereas the orders Agaricales, Tremellales and Chaetothyriales showed low sensitivity to seasonality (Fig. 4c). Regarding the timing of the seasonal peak, we did not observe any phylogenetic signal (Pagel's $\lambda$ = −0.04, $P$ = 0.80). However, this lack of a signal may be partially explained by the fact that few species showed sufficient seasonality for the time of the optimal season to be defined (Fig. 4a).

## Discussion

Our results show that fungi follow predictable latitudinal diversity gradients that resemble other major groups of organisms[48]. This finding represents a major contribution to the long-standing debate over whether organisms with a microbial lifestyle follow the global biodiversity paradigms known for macroorganisms[4,5]. Our results are consistent with an increasing body of literature showing that, like macroorganisms, microbial communities are spatially structured at large scales[6,7,16]. Interestingly, only a small minority of all species-level OTUs detected in our study were observed in all three climatic zones. These widespread species were Ascomycota genera that have previously been found to be very common in both soil[49] and air[17]. However, the vast majority of OTUs were detected only within one climatic zone and the spatiotemporal patterns of species richness and community composition were highly constrained by climatic conditions. Although previous large-scale studies of soil fungi have found clear effects of climate on community composition[16,19], the fact that in our data MAT explains most of the variation in the distributions of fungi is striking, especially given that our data are based on the dispersal stage of airborne spores. Likewise, previous studies on soil fungal communities have found that biomes, as defined based on MAT and MAP, explain a major part of their global distribution[16].

A major advantage of our data is the high level of temporal replication, enabling a global analysis of climatic effects on the phenology of fungal reproduction. Seasonality in both the amount of DNA and species richness of airborne fungi increased with increasing distance from the Equator and therefore seasonality was highest in Arctic climates. Less trivially, we found that seasonal turnover in community composition increased with increasing distance from the Equator, even if tropical regions also show high seasonality (for example, rainy versus dry periods). In line with this finding, a long-term study of airborne fungi in the tropics showed no seasonality[29]. In addition to seasonal effects, our study also highlights the importance of short-term local weather conditions on the diversity or sporulation phenology of airborne fungi. The results showed that airborne fungal species richness peaks during warm and windy sampling days, a finding coinciding with previous observations that temperature influences fungal reproductive phenology[32] and that spore release peaks when wind speeds are high[50].

Comparison of trophic guilds showed that not only overall species richness, but also most guilds, were most abundant in the tropics, with the notable exceptions of lichenized, ericoid mycorrhizal and ectomycorrhizal species. This result is in line with the patterns demonstrated for soil fungi by Tedersoo et al.[16], who also found a general increase towards the tropics, except for ectomycorrhizal fungi which were most diverse in boreal and temperate regions. Whereas the higher diversity of these fungal groups at higher latitudes could be related to greater knowledge gaps of their diversity in the tropics, this result could also reflect the distribution and diversity of their host species[51]. To minimize the possibility of misleading artefact due to knowledge gaps, we borrowed information among taxonomic levels for the functional classifications, making a compromise between minimization of bias (by inclusion of not only the minority of OTUs reliably classified to species but also genus- or family-level classifications) and minimization of the noise of false classifications (by not borrowing information from ranks higher than family). In terms of seasonality, many earlier studies have reported longer sporulation and reproductive seasons in warmer regions for specific parts of the world and for particular groups of fungi[32,52]. Our results generalize these earlier findings to the global distribution of the entire fungal kingdom: all fungal guilds showed consistent and predictable patterns, with sporulation activity being shorter and more pronounced towards higher latitudes. Regarding spore size, we found that asexual spore size decreased but sexual spore size increased with increasing distance from the Equator. During the main reproductive season in the temperate and polar-continental zones, we further found asexual spore size to increase but sexual spore size to decrease during the season. The latter result, which is consistent with the earlier finding of Kauserud et al.[53], is partially generated by ascomycetes having on average larger sexual spores[54] and earlier sporulation phenology than basidiomycetes[33]. Our study reports opposing spatial and temporal patterns between sexual and asexual spores, suggesting contrasting evolutionary forces behind the size of these two types of dispersal propagule. This result may also relate to the opposing environmental

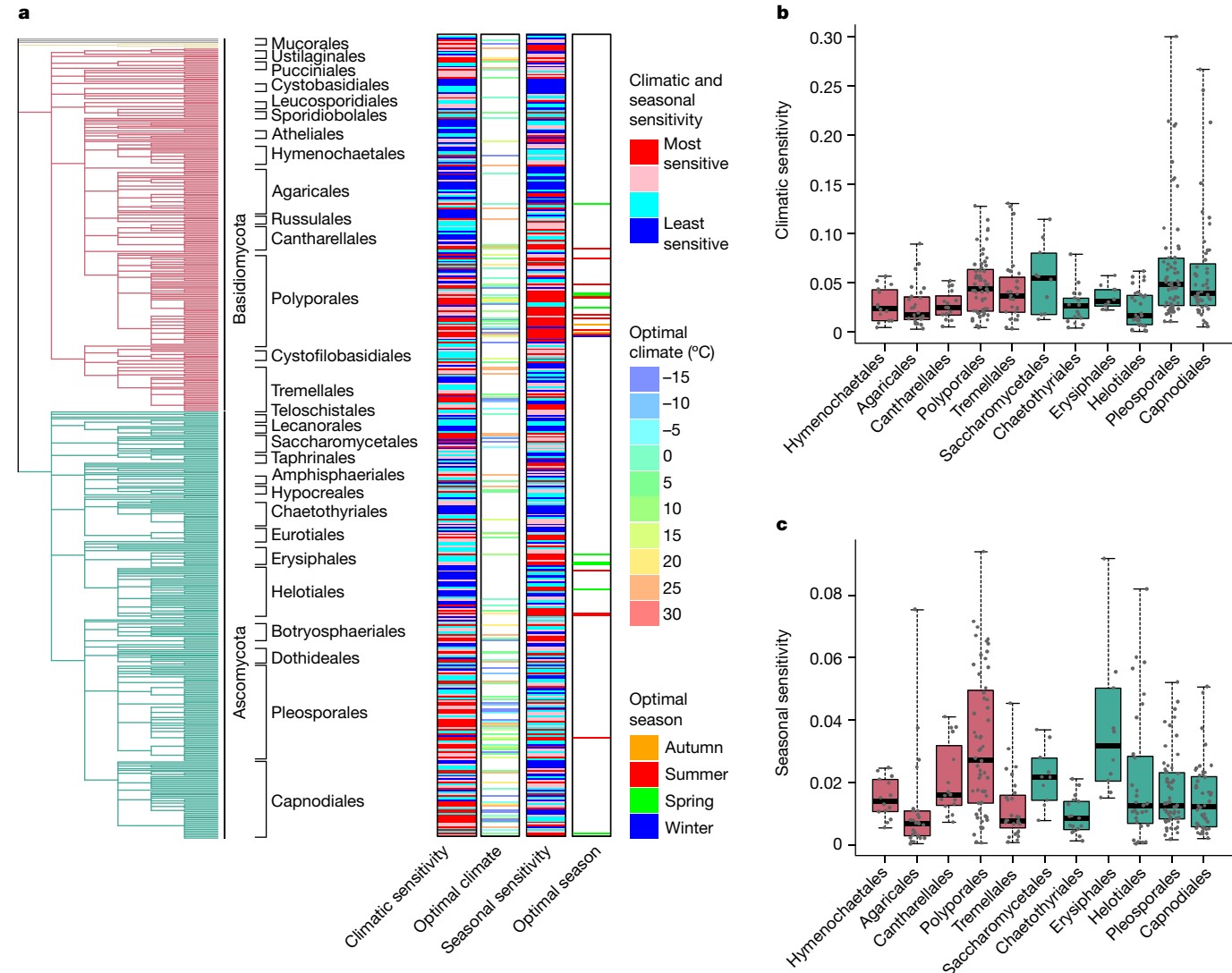

**Fig. 4 | Phylogenetic signal in climatic and seasonal variation. a–c,** All results are based on a joint-species distribution model fitted to the 485 most common species. **a,** Quantification of variation in climatic sensitivity, optimal climate, seasonality sensitivity and optimal season among species. For climatic and seasonal sensitivity the colours show the proportion of variance explained by the second-order polynomial of the MAT of that site (for climatic sensitivity) and by the periodic functions of $\sin(2\pi d/365)$ and $\cos(2\pi d/365)$, where $d$ represents the Julian day of the year (for seasonal sensitivity), coded as blue, cyan, pink and red for the four quartiles. For optimal climate we show the MAT at which the second-order polynomial of that MAT was maximized (that is, the point at which a further increase in MAT will change an estimated increase to an estimated decrease in species occurrence) in the colour scale of the world map shown in Fig. 1a. For optimal season we show the season at

which the estimated occurrence of the species will peak, with colours coded as blue for winter (December–February in the Northern Hemisphere; for the Southern Hemisphere we assumed a 6 month difference in seasonality), green for spring (March–May), red for summer (June–August) and orange for autumn (September–November). Cases in which climatic or seasonal sensitivities were too low to determine the optimal climate or season are shown in white. **b,c,** Boxplots show the distributions of climatic (**b**) and seasonal sensitivities (**c**) for those orders represented in these analyses by at least ten species. Lines show the medians, boxes the lower and upper quartiles and whiskers the minimum and maximum values. The raw data are shown by dots that have been jittered to show overlapping points. For the list of taxa included in the analysis, see Supplementary Information.

triggers of sexual and asexual spore production, with the former occurring especially under unfavourable environmental conditions such as at the end of the growing season[55,56].

In terms of the processes that structure ecological communities, we may distinguish between (1) the ultimate evolutionary processes that give rise to species and determine their traits and (2) the proximate contemporary ecological processes that shape the assembly of communities[57,58]. Our data on global aerial communities shed light on both aspects. In terms of evolutionary processes, fungi exhibited a strong niche conservatism regarding sensitivity to dispersal seasonality and moderate conservatism for sensitivity to climatic conditions. These results suggest that fungi have continuously adapted to climatic

conditions rather than being stuck in their ancestral climatic niches. This interpretation is supported by the fact that whereas most species showed climatically restricted distributions, the majority of genera and the vast majority of orders were detected in all three climatic zones. The high phylogenetic signal in dispersal seasonality was driven by certain taxonomic groups. In particular, Polyporales showed a high level of seasonality for almost all species. Our findings suggest that Polyporales have been especially adapted to seasonal climates, possibly because their morphological and physiological traits support high spore production for a brief portion of the fruiting season. Among the ecological selection processes, we showed that environmental drivers, in particular MAT, play a major structuring role in fungal communities at large scales.

Whereas substrate-specific sampling will mainly show the DNA of mycelia locally present in the focal substrates, aerial DNA will provide an integrated view of airborne propagules from all substrates. As evidence, all trophic guilds supported by the guild database we used are represented in the data. However, some functional groups were better represented than others, highlighting the importance of surveying different complementary substrates to gain a complete view of fungal diversity. Importantly, the proportional representation of aerial fungal taxa is clearly affected by their dispersal strategy: in particular, plant pathogens, saprotrophs and wood saprotrophs were very abundant in our data (Fig. 2a). By contrast, ectomycorrhizal fungi, not all of which produce conspicuous and abundant above-ground reproductive bodies, contributed only a small fraction of airborne spores globally (Fig. 2a). This points to other dispersal means—for example, via mycophagous animals—as being important for this functional group. Alternatively, the relative scarcity of airborne spores from ectomycorrhizal fungi may be due to the trade-off between spore size and number[45]. Because many ectomycorrhizal fungi develop large spores they are expected to produce fewer spores, which in turn would appear less frequently in airborne data. Note that typically both large and small spores are unicellular and contain a single nucleus.

Our results demonstrate that the sampling of airborne DNA can provide a synthetic, cumulative view of global fungal diversity across individual substrates. This integrated view provides a huge step forward in the understanding of the distributions and dynamics of the whole fungal kingdom, which has lagged behind research in other major organism groups partially due to methodological difficulties in surveying fungi comprehensively. Overall our results show highly predictable patterns of spatial and seasonal variation in airborne fungi and suggest that the drivers of microbial community assembly are largely similar to those determining the assembly of macroorganisms. Our results highlight the role of temperature as an underlying driver of fungal dynamics, with fungal diversity increasing with warmer climates and sporulation activity increasing with warmer days. This finding suggests that global climate change with generally warming climates will have a major role in restructuring fungal communities.

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

Nerea Abrego[1,2,90 ✉], Brendan Furneaux[1,90], Bess Hardwick[2], Panu Somervuo[3], Isabella Palorinne[2], Carlos A. Aguilar-Trigueros[1], Nigel R. Andrew[4,5], Ulyana V. Babiy[6], Tan Bao[7], Gisela Bazzano[8], Svetlana N. Bondarchuk[9], Timothy C. Bonebrake[10], Georgina L. Brennan[11], Syndonia Bret-Harte[12], Claus Bässler[13,14,15], Luciano Cagnolo[16], Erin K. Cameron[17], Elodie Chapurlat[18], Simon Creer[19], Luigi P. D'Acqui[20], Natasha de Vere[21], Marie-Laure Desprez-Loustau[22], Michel A. K. Dongmo[10,23], Ida B. Dyrholm Jacobsen[24], Brian L. Fisher[25,26], Miguel Flores de Jesus[27], Gregory S. Gilbert[28], Gareth W. Griffith[29], Anna G. Gritsuk[9], Andrin Gross[30], Håkan Grudd[31], Panu Halme[1], Rachid Hanna[32], Jannik Hansen[33], Lars Holst Hansen[33], Apollon D. M. T. Hegbe[34], Sarah Hill[4], Ian D. Hogg[35,36,37], Jenni Hultman[38,39], Kevin D. Hyde[40], Nicole A. Hynson[41], Natalia Ivanova[42,43], Petteri Karisto[44,45], Deirdre Kerdraon[18], Anastasia Knorre[46,47], Irmgard Krisai-Greilhuber[48], Juri Kurhinen[3], Masha Kuzmina[42], Nicolas Lecomte[49], Erin Lecomte[49], Viviana Loaiza[50], Erik Lundin[31], Alexander Meire[31], Armin Mešić[51], Otto Miettinen[52], Norman Monkhouse[42], Peter Mortimer[53], Jörg Müller[14,54], R. Henrik Nilsson[55], Puani Yannick C. Nonti[34], Jenni Nordén[56], Björn Nordén[56], Veera Norros[57], Claudia Paz[58,59], Petri Pellikka[60,61,62], Danilo Pereira[44,63], Geoff Petch[64], Juha-Matti Pitkänen[39], Flavius Popa[65], Caitlin Potter[29], Jenna Purhonen[1,66], Sanna Pätsi[67], Abdullah Rafiq[19], Dimby Raharinjanahary[26], Niklas Rakos[31], Achala R. Rathnayaka[40,68], Katrine Raundrup[24], Yury A. Rebriev[69], Jouko Rikkinen[3,52], Hanna M. K. Rogers[18], Andrey Rogovsky[46], Yuri Rozhkov[70], Kadri Runnel[71,72], Annika Saarto[67], Anton Savchenko[72], Markus Schlegel[30], Niels Martin Schmidt[33,73], Sebastian Seibold[74,75], Carsten Skjøth[64,76], Elisa Stengel[77], Svetlana V. Sutyrina[9], Ilkka Syvänperä[78], Leho Tedersoo[71,79], Jebidiah Timm[12], Laura Tipton[80], Hirokazu Toju[81,82], Maria Uscka-Perzanowska[18], Michelle van der Bank[83], F. Herman van der Bank[83], Bryan Vandenbrink[35], Stefano Ventura[20], Solvi R. Vignisson[84], Xiaoyang Wang[85], Wolfgang W. Weisser[75], Subodini N. Wijesinghe[40,68], S. Joseph Wright[86], Chunyan Yang[85], Nourou S. Yorou[34], Amanda Young[12], Douglas W. Yu[85,87,88], Evgeny V. Zakharov[36,42], Paul D. N. Hebert[36,42], Tomas Roslin[3,18] & Otso Ovaskainen[1,3,89]

[1]Department of Biological and Environmental Science, University of Jyväskylä, Jyväskylä, Finland. [2]Department of Agricultural Sciences, University of Helsinki, Helsinki, Finland. [3]Organismal and Evolutionary Biology Research Programme, Faculty of Biological and Environmental Sciences, University of Helsinki, Helsinki, Finland. [4]Natural History Museum, University of New England, Armidale, New South Wales, Australia. [5]Faculty of Science and Engineering, Southern Cross University, Northern Rivers, New South Wales, Australia. [6]Wrangel Island State Nature Reserve, Pevek, Russia. [7]Department of Biological Sciences, MacEwan University, Edmonton, Alberta, Canada. [8]Centro de Zoología Aplicada, Facultad de Ciencias Exactas Físicas y Naturales, Universidad Nacional de Córdoba, Córdoba, Argentina. [9]Sikhote-Alin State Nature Biosphere Reserve named after K. G. Abramov, Terney, Russia. [10]School of Biological Sciences, The University of Hong Kong, Hong Kong SAR, China. [11]Institute of Marine Sciences, Consejo Superior de Investigaciones Científicas (CSIC), Passeig Marítim de la Barceloneta, Barcelona, Spain. [12]Institute of Arctic Biology, University of Alaska, Fairbanks, AK, USA. [13]Department of Conservation Biology, Institute for Ecology, Evolution and Diversity, Faculty of Biological Sciences, Goethe-University Frankfurt, Frankfurt am Main, Germany. [14]Bavarian Forest National Park, Grafenau, Germany. [15]Bayreuth Center of Ecology and Environmental Research (BayCEER), University of Bayreuth, Bayreuth, Germany. [16]Instituto Multidisciplinario de Biología Vegetal, Consejo Nacional de Investigaciones Científicas y Técnicas (CONICET), Córdoba, Argentina. [17]Department of Environmental Science, Saint Mary's University, Halifax, Nova Scotia, Canada. [18]Department of Ecology, Swedish University of Agricultural Sciences (SLU), Uppsala, Sweden. [19]Molecular Ecology and Evolution at Bangor (MEEB), School of Biological Sciences, Bangor University, Bangor, Wales. [20]Research Institute on Terrestrial Ecosystems - IRET, National Research Council - CNR and National Biodiversity Future Center, Palermo, Italy. [21]Natural History Museum of Denmark, University of Copenhagen, Copenhagen, Denmark. [22]BIOGECO, INRAE, University of Bordeaux, Cestas, France. [23]International Institute of Tropical Agriculture (IITA), Yaoundé, Cameroon. [24]Greenland Institute of Natural Resources, Nuuk, Greenland. [25]Department of Entomology, California Academy of Sciences, San Francisco, CA, USA. [26]Madagascar Biodiversity Center, Parc Botanique et Zoologique de Tsimbazaza, Antananarivo, Madagascar. [27]Legado das Águas, Reserva Votorantin, Tapiraí, Brazil. [28]Department of Environmental Studies, University of California Santa Cruz, Santa Cruz, CA, USA. [29]Department of Life Sciences, Aberystwyth University, Aberystwyth, UK. [30]Biodiversity and Conservation Biology Research Unit, SwissFungi Data Center, Swiss Federal Research Institute WSL, Birmensdorf, Switzerland. [31]Swedish Polar Research Secretariat, Abisko Scientific Research Station, Abisko, Sweden. [32]Center for Tropical Research, Congo Basin Institute, University of California Los Angeles, Los Angeles, CA, USA. [33]Department of Ecoscience, Aarhus University, Roskilde, Denmark. [34]Research Unit in Tropical Mycology and Plant–Soil Fungi Interactions, Faculty of Agronomy, University of Parakou, Parakou, Republic of Benin. [35]Canadian High Arctic Research Station, Polar Knowledge Canada, Cambridge Bay, Nunavut, Canada. [36]Department of Integrative Biology, College of Biological Science, University of Guelph, Guelph, Ontario, Canada. [37]School of Science, University of Waikato, Hamilton, New Zealand. [38]Department of Microbiology, University of Helsinki, Helsinki, Finland. [39]Natural Resources Institute Finland (Luke), Helsinki, Finland. [40]Center of Excellence in Fungal Research, Mae Fah Luang University, Chiang Rai, Thailand. [41]Pacific Biosciences Research Center, University of Hawaii at Manoa, Honolulu, HI, USA. [42]Centre for Biodiversity Genomics, University of Guelph, Guelph, Ontario, Canada. [43]Nature Metrics North America Ltd., Guelph, Ontario, Canada. [44]Plant Pathology Group, Institute of Integrative Biology, ETH Zurich, Zurich, Switzerland. [45]Plant Health, Natural Resources Institute Finland (Luke), Jokioinen, Finland. [46]Science Department, National Park Krasnoyarsk Stolby, Krasnoyarsk, Russia. [47]Institute of Ecology and Geography, Siberian Federal University, Krasnoyarsk, Russia. [48]Department of Botany and Biodiversity Research, University of Vienna, Vienna, Austria. [49]Centre d'Études Nordiques and Canada Research Chair in Polar and Boreal Ecology, Department of Biology, Université de Moncton, Moncton, New Brunswick, Canada. [50]Department of Evolutionary Biology and Environmental Sciences, University of Zürich, Zurich, Switzerland. [51]Laboratory for Biological Diversity, Rudjer Boskovic Institute, Zagreb, Croatia. [52]Finnish Museum of Natural History, University of Helsinki, Helsinki, Finland. [53]Centre for Mountain Futures, Kunming Institute of Botany, Chinese Academy of Sciences, Kunming, China. [54]Department of Conservation Biology and Forest Ecology, Julius Maximilians University Würzburg, Rauhenebrach, Germany. [55]Department of Biological and Environmental Sciences, Gothenburg Global Biodiversity Centre, University of Gothenburg, Gothenburg, Sweden. [56]Norwegian Institute for Nature Research (NINA), Oslo, Norway. [57]Nature Solutions, Finnish Environment Institute (Syke), Helsinki, Finland. [58]Department of Biodiversity, Institute of Biosciences, São Paulo State University, Rio Claro, Brazil. [59]Department of Entomology and Acarology, Laboratory of Pathology and Microbial Control, University of São Paulo, Piracicaba, Brazil. [60]Department of Geosciences and Geography, Faculty of Science, University of Helsinki, Helsinki, Finland. [61]State Key Laboratory for Information Engineering in Surveying, Mapping and Remote Sensing, Wuhan University, Wuhan, China. [62]Wangari Maathai Institute for Environmental and Peace Studies, University of Nairobi, Kangemi, Kenya. [63]Laboratory of Biochemistry, Wageningen University, Wageningen, the Netherlands. [64]School of Science and the Environment, University of Worcester, Worcester, UK. [65]Department of Ecosystem Monitoring, Research & Conservation, Black Forest National Park, Bad Peterstal-Griesbach, Germany. [66]School of Resource Wisdom, University of Jyväskylä, Jyväskylä, Finland. [67]Biodiversity Unit, University of Turku, Turku, Finland. [68]School of Science, Mae Fah Luang University, Chiang Rai, Thailand. [69]Southern Scientific Center of the Russian Academy of Sciences, Rostov-on-Don, Russia. [70]State Nature Reserve Olekminsky, Olekminsk, Russia. [71]Mycology and Microbiology Center, University of Tartu, Tartu, Estonia. [72]Institute of Ecology and Earth Sciences, University of Tartu, Tartu, Estonia. [73]Arctic Research Center, Aarhus University, Roskilde, Denmark. [74]Forest Zoology, TUD Dresden University of Technology, Berchtesgaden, Germany. [75]Terrestrial Ecology Research Group, Department of Life Science Systems, School of Life Sciences, Technical University of Munich, Freising, Germany. [76]Department of Environmental Science, Aarhus University, Roskilde, Denmark. [77]Field Station Fabrikschleichach, Department of Animal Ecology and Tropical Biology (Zoology III), Julius Maximilians University Würzburg, Rauhenebrach, Germany. [78]Kevo Subarctic Research Institute, Biodiversity Unit, University of Turku, Utsjoki, Finland. [79]College of Science, King Saud University, Riyadh, Saudi Arabia. [80]School of Natural Science and Mathematics, Chaminade University of Honolulu, Honolulu, HI, USA. [81]Laboratory of Ecosystems and Coevolution, Graduate School of Biostudies, Kyoto University, Kyoto, Japan. [82]Center for Living Systems Information Science (CeLiSIS), Graduate School of Biostudies, Kyoto University, Kyoto, Japan. [83]African Centre for DNA Barcoding (ACDB), University of Johannesburg, Auckland Park, South Africa. [84]Sudurnes Science and Learning Center, Sandgerði, Iceland. [85]State Key Laboratory of Genetic Resources and Evolution, Kunming Institute of Zoology, Chinese Academy of Sciences, Kunming, China. [86]Smithsonian Tropical Research Institute, Balboa, Panama. [87]School of Biological Sciences, University of East Anglia, Norwich, UK. [88]Yunnan Key Laboratory of Biodiversity and Ecological Security of Gaoligong Mountain, Kunming Institute of Zoology, Center of Excellence in Animal Evolution and Genetics, Chinese Academy of Sciences, Kunming, China. [89]Centre for Biodiversity Dynamics, Department of Biology, Norwegian University of Science and Technology, Trondheim, Norway. [90]These authors contributed equally: Nerea Abrego, Brendan Furneaux. ✉e-mail: nerea.n.abrego-antia@jyu.fi

## Methods

### Sampling, sequencing and bioinformatics

For full details on study design and sample collection, DNA extraction and sequencing, bioinformatic processing, as well as technical data validation, see ref. 3. Here we summarize these steps.

The study design consists of 47 sampling sites, each equipped with a cyclone sampler (Burkard Cyclone Sampler for Field Operation, Burkard Manufacturing Co Ltd; http://burkard.co.uk/product/cyclone-sampler-for-field-operation). The sampling sites were selected to represent local natural environments in which intensive, continuous sampling was possible. The cyclone samplers collected particles of greater than 1 μm in size from the air directly into a sterile Eppendorf vial, with average air throughput of 23.8 m³ during each 24 h sampling period. Before the start of our global sampling, a field test was performed to evaluate the quantity of fungal DNA collected over different time frames. We also included field blanks handled with and without gloves, in which the sampler was not activated, and the Eppendorf vials were removed after 1 min and sealed. As a result of the field tests we selected a 24 h sampling period and instructed the participants to handle samples with gloves and to clean the cyclone parts monthly.

We amplified the ITS2 region using PCR for 20 cycles with fusion primers ITS_S2F[60], ITS3 and ITS4 (ref. 61) tailed with Illumina adaptors and sequenced them on Illumina MiSeq. In the MiSeq runs we included two sets of negative control samples, introduced at the DNA extraction step and the PCR step, respectively. Of the 99 negative control samples, 89% (88 samples) did not yield any reads of fungal origin. The remaining nine negative control samples included a few fungal reads (relative to the study samples) of relatively common OTUs, suggesting infrequent cross-contamination. To test the robustness of the results with respect to such cross-contamination, we repeated three of the main analyses (variation in overall species richness, variation in guild-specific species richness and joint-species distribution modelling) with data that we purposely contaminated with the observed level of cross-contamination. To do so we added to the OTU reads of each field sample the OTU reads of a randomly selected negative control sample. We replicated the cross-contamination simulation for ten independent replicates, with the results being almost identical to those obtained from the original data (Supplementary Information and Extended Data Figs. 4–6). To quantify the amount of fungal DNA we applied a spike-in approach and converted the ratio of non-spike versus spike sequences into semiquantitative estimates of DNA amount[35]. Demultiplexed paired-end reads were trimmed, denoised and chimera checked using Cutadapt v.4.2 (ref. 62), DADA2 v.1.18.0 (ref. 63) and VSEARCH v.2.22.1 (ref. 64). As a reference database we used Sanger sequences from the UNITE v.9 database[65] supplemented with the synthetic spike sequences. Sequences representing non-spike ASVs[38] were aligned between the ITS3 and ITS4 primer sites. Discarding of sequences that did not match the full length of the model, or with a bit score less than 50, resulted in a 65,912 ASV × 2,768 sample matrix of read abundance.

Due to the unsuitability of using ITS-based ASVs as proxies for species[66], we developed a taxonomically guided clustering approach to form species-level OTUs. We performed a probabilistic taxonomic placement of ASVs with Protax-fungi[40] with a 90% probability threshold. In addition, sequences whose best match to UNITE Sanger sequences was to a kingdom other than Fungi were annotated as potential non-fungi. We applied constrained clustering by first forming cluster cores by those ASVs that had been assigned to taxa by Protax-fungi. We then matched unassigned ASVs to the closest cluster core using optimized sequence similarity thresholds. Finally, remaining unclustered ASVs were clustered using de novo single-linkage clustering. These de novo clusters were assigned to placeholder taxonomic names of the form 'pseudo{rank}_{number}'. The final result of this process was

a read abundance matrix of 27,954 species-level OTUs × 2,768 samples, along with taxonomic annotations at each rank from phylum to species, including pseudotaxon placeholders.

The mean sequencing depth (total number of fungal and spike sequences) among samples was 86,845 sequences per sample. Based on rarefaction analyses presented in ref. 3 we discarded samples that did not contain at least 10,000 sequencing reads, representing 1.8% of samples. To avoid losing some OTUs detected in the most diverse samples, we controlled for variation in sequencing depth by statistical means rather than using rarefied values[67].

### Weather and climate data

Weather variables were extracted from ERA5 hourly data on single levels dataset[42] available at the Copernicus Climate Data Store (https://cds.climate.copernicus.eu/cdsapp#!/home). To download weather variables we used the R package ecmwfr[68]. We downloaded hourly data on (1) 2m_temperature—that is, instantaneous temperature (k) at 2 m height (henceforth termed temperature), (2) total_precipitation—that is, precipitation (m) accumulated over a 1 h period (henceforth termed precipitation), (3) 10m_v_component_of_wind—that is, horizontal speed (m s⁻¹) of air moving towards the north at a height of 10 m and (4) 10m_u_component_of_wind—that is, horizontal speed (m s⁻¹) of air moving towards the east at a height of 10 m. The latter two variables (wind to north $v$ and wind to east $u$) were combined to compute wind speed by applying the formula $\sqrt{v^2 + u^2}$. All four variables were downloaded for the latitude range from −80 to 80 degrees and longitude range from −180 to 180 degrees for the period 7 May 2018 to 2 February 2021, which extended well past our study period. We then averaged the hourly data to daily data and extracted data for the sampling locations of our study. We downloaded climatic data using the same tools but with the 'sis biodiversity ERA5 global dataset'. As climatic variables we included the 40-year averages (1979–2018) of annual_mean_temperature (MAT), annual_precipitation (MAP), wind_speed and aridity (MAI).

### Extraction of spore size and trophic guild data

We extracted spore size and trophic guild data from the data assembled by Aguilar-Trigueros et al.[54]. Spore size data originate from species-level taxonomic descriptions in Mycobank[69] (containing spore dimension data for over 36,000 species) and include, for every fungal species, the sizes of spores produced in both sexual and asexual cycles. The trophic guild data consist of a compilation of recordings of fungal functions across major databases (see ref. 54 for a detailed list of compiled databases).

Connecting spore volume data to molecularly identified species is not straightforward, because (1) some taxa were identified only to a higher taxonomic level than species and (2) the spore volume databases are not complete. For those OTUs identified to species level and for which a spore volume estimate was available we used the species-level estimate. When a species-level estimate was not available we used the genus-level estimate, computed as the average over the species belonging to the focal genus. When a genus-level estimate was not available we used the family-level estimate, computed as the average over the genera belonging to the focal family. If a family-level estimate was not available we considered the spore volume for the focal species as missing data. We computed the community-weighted mean (CWM) of log-transformed spore volume for each sample as the average log-transformed spore volume over the species present in the sample. When doing so we distinguished between spores produced during asexual (that is, asexual spores) and sexual cycles (that is, sexual spores), thus resulting in CWM sizes of both asexual and sexual spores. We note that this analysis is based on the molecular classifications of the ITS2 sequences rather than, for example, direct microscopy of the sampled spores, and hence we cannot distinguish whether the spores in the samples were asexual or sexual. Therefore, these variables should

be interpreted as the mean size of the asexual or sexual spores of those species present in the sample.

When assigning the trophic guild data we included only the most common trophic guilds and grouped some of them (Extended Data Table 2). We first matched those OTUs identified to the species level and which matched a species in the database of Aguilar-Trigueros et al.[54]. In those cases for which an OTU was identified to only genus level, or species-level identification was not available in the database, we assigned from the database all trophic guild categories of the species belonging to the focal genus; likewise, when the OTU was identified only to the family level we assigned from the database all trophic guild categories of the species belonging to the focal family. As result, some OTUs were assigned to more than one trophic guild and hence the classifications should be considered as potential guilds to which the OTU may belong, often based on information borrowed from its relatives.

## Variation in community composition

We conducted multivariate analyses at the site, rather than at the sample, level. For each site we measured the abundance of each taxon by its prevalence—that is, the proportion of samples in which it was present. We then computed the site-to-site community distance matrix using either the Bray–Curtis dissimilarity index (using the vegdist function of the R package vegan[70]) or, alternatively, the unifrac distance (using the UniFrac function of the R package phyloseq[71]) that accounted for taxonomic relatedness among the taxa. As candidate environmental variables used to explain community dissimilarity we used MAT, MAP, MAI and mean annual wind speed, all averaged over the 40-year period from 1979 to 2018. The reason for including only a small number of site-specific variables in the analysis is that, whereas the study is global in scope, it includes only 47 sites. The data thus hold limited information on statistically disentangling the effects of many spatially varying covariates. Instead, the main strength of the study lies in its high temporal replication, which allowed us to identify effects of the spatiotemporal covariates such as seasonality.

We visualized the community distance matrices with non-metric multidimensional scaling (NMDS; using the metaMDS function of the R package vegan) and illustrated the effect of each candidate environmental variable on the ordination space (using the ordisurf function of the R package vegan). To partition the variation in community dissimilarity explained by spatial distance and by each candidate environmental variable, we used linear models in which community dissimilarity was explained by either geographic distance, environmental distance or both. We computed the proportions of variance explained by space alone, by environment alone and by shared effect following Whittaker[72].

## Univariate analyses addressing how variations in DNA amount, species richness, spore size and trophic guild composition depend on climate, season and weather

We fitted a series of mixed linear models with the R package lme4 (ref. 73) for each of the following response variables: log(DNA amount), log(species richness + 1), CWM log(sexual spore size), CWM log(asexual spore size) and log(number of species classified to each trophic guild + 1). For analyses concerning spore size we included only samples that contained at least ten species, to reduce noise in the response variables. In addition to conducting the analyses for CWM computed for all species, we also repeated the spore size analyses with restriction for basidiomycetes only and for ascomycetes only. These additional analyses were motivated by the question of whether the results were consistent among these two major groups.

As described in greater detail below, we considered four models (CS1–CS4) of climatic and seasonal variation. In addition to the best-supported model of climatic and seasonal variation we considered four models (W1–W4) of weather variation, each of which further consisted

of 64 variants according to which weather variables they included. We describe these model variants below and illustrate them conceptually in Supplementary Information and Extended Data Fig. 7. We performed model selection among these model variants with Akaike information criterion (AIC) and used the explanatory powers of the models to assess the proportion of total variation they explain.

**Influence of climatic and seasonal variation.** To evaluate the effects of climatic and seasonal variation we considered the following four nested models, described in order of increasing complexity.
1. Model CS1: null model. The null model does not include any ecological predictors as fixed effects but includes log(sequencing depth) for the species richness model. To account for the study design with multiple samples from the same locations, the null model includes the site as a random intercept.
2. Model CS2: climate dependence. In this model we assumed that the response variable varies systematically with the MAT of the site. Thus, we extended model CS1 by including a fixed effect of MAT and its square.
3. Model CS3: climate dependence and latitude-dependent seasonality. In this model we assumed that the response variable additionally shows seasonal variation that systematically depends on latitude. We thus extended model CS2 by including as fixed effects the interaction between latitude and seasonality. We modelled seasonality with the periodic functions $\sin\left(2\frac{\pi d}{365}\right)$ and $\cos\left(2\frac{\pi d}{365}\right)$, where $d$ is the Julian day of the year. Because latitude is positive for the Northern and negative for the Southern Hemisphere, we note that the interaction between seasonality and latitude assumes opposite patterns of seasonality in the two hemispheres. It is thus appropriate to account for the 6 month difference in seasonality between the two hemispheres.
4. Model CS4: climate dependence and site-specific seasonality. Model CS4 extends model CS3 by including the random effect of the site not only in the intercept, but also as random slopes related to seasonality. This model thus assumes that each site may show a deviation from the systematic latitude-dependent variation in seasonality, generated by some site-specific effects not included in the model.

**Influence of weather variation.** The aim of these analyses was to assess how the prevailing weather conditions influence the four response variables. As weather-related covariates we used temperature, precipitation and wind speed and added these covariates as additional predictors to CS4, the most complex climatic model. Because weather variables (especially temperature) follow seasonal patterns that depend on latitude, using them as such would confound their effects with those of the climatic and seasonal predictors. For this reason we included the covariates as the difference between the actual values and those expected based on latitude and season; henceforth we term these temperature, precipitation and wind-speed anomalies. We calculated these anomalies as the differences between the daily observed values and the predictions of site-specific seasonality models (that is, model CS4) fitted to each weather covariate. For example, the temperature anomaly for a given day and site describes how much warmer (positive anomaly) or colder (negative anomaly) that site was compared with what would be expected for that site in that season. Furthermore, we note that the weather covariates may influence variation in fungal communities through either their effect on detection (for example, prevailing wind conditions during sampling) or their influence on production of fruiting bodies and sporulation (for example, temperature and humidity conditions over the past week). Because the timescales at which climatic conditions influence spore production are generally unknown and can vary among species, we computed the weather predictors in three alternative ways, averaging them over a period of either 1 day, 1 week or 1 month before sampling. We considered the full set of candidate models in which each weather covariate was either excluded

or included at the time scale of day, week or month. Because there are three weather covariates and each of them has four options the number of candidate models is 64, encompassing the null model in which no weather covariates were included. In regard to how we assumed that weather would influence the response variables we considered the following four nested models, each of which included as baseline the best-supported model of climate and seasonality.

1. Model W1: constant weather effects. Model W1 includes in the fixed effects the main effects of weather covariates.
2. Model W2: weather effects depend on the site. Model W2 extends model W1 by also including in the fixed effects the interactions between climatic variables (MAT and its square) and weather covariates, as well as in the random effects the interactions between site and weather covariates, thus allowing temperature anomaly to have a site-specific effect that potentially varies systematically with climate.
3. Model W3: weather effects depend on both the site and latitude-dependent seasonality. Model W3 extends model W2 by also featuring inclusion in the fixed effects the interactions between latitude-dependent seasonality (the interaction between latitude and periodic functions of the day of the year) and weather covariates, thus allowing, for example, temperature anomaly to have a positive effect in spring but negative effect in autumn.
4. Model W4: weather effects depend on both the site and site–dependent seasonality. Model W4 extends model W3 by including in the random effects the effect of the site, and the slopes related to interaction between seasonality and the weather covariates. This model thus assumes that the effects of the weather covariates show site-specific variation in both their mean effect and seasonality.

### Seasonality in community composition

To characterize how seasonality in community composition is dependent on climate we computed for each site an index of seasonality in community composition and then fitted a linear model in which we regressed this index against the MAT of the site. To describe seasonality in community composition we examined how much more similar pairs of samples were in terms of their community composition if they were sampled from the same season compared with whether they were sampled from different seasons. We considered a pair of samples as belonging to the same season if they were taken at most 1 month apart, whereas we considered them as belonging to a different season if they were taken 3 months (plus or minus half a month) from each other. As a measure of community similarity we used the Jaccard similarity index, which we averaged over those pairs of samples that contained at least five species. We then used an index of seasonality in community composition calculated as the average Jaccard similarity index for pairs of samples that were taken in the same season, minus the average Jaccard similarity index for pairs of samples taken in a different season. We accounted for the Jaccard similarity index for pairs of samples that were taken in the same season to control for possible variation in the baseline turnover and thus to extract the sole effect of seasonality.

### Joint-species distribution modelling of phylogenetic signal in climatic and seasonal variation

To examine for phylogenetic signals in climatic and seasonal variation we analysed the data with HMSC[47,74]. HMSC is a joint-species distribution model[75] that includes a hierarchical layer modelling how species' environmental covariates relate to their traits and/or phylogenetic relationships[76]. We restricted these analyses to the 485 species that occurred in the data at least 50 times, and therefore had sufficient data to estimate climatic and seasonal responses. As the response variable we used the presence/absence of species at the level of the sample, which we modelled through the Bernoulli distribution and probit-link function. To measure climatic responses we included as

fixed effects the second-order polynomial of the MAT of the site. To measure seasonal responses we also included as fixed effects the interaction between latitude and seasonality that we modelled with the periodic functions $\sin(2\pi d/365)$ and $\cos(2\pi d/365)$, where $d$ is the Julian day of the year. To control for variation in sequencing depth (that is, the number of sequences obtained for each sample) we also included the log-transformed sequencing depth as fixed effect. To control for repeated samples from the same sites we included the site as a random effect. To examine how species' responses to the predictors related to their phylogenetic relationships we included in the HMSC model a taxonomic tree in which we assumed equal branch lengths at the levels of phylum, class, order, family, genus and species.

We fitted the model with the R package HMSC[77] assuming the default prior distributions[47]. We sampled the posterior distribution with four Markov chain Monte Carlo chains, each of which was run for 37,500 iterations of which the first 12,500 were removed as burn-in. The chains were thinned by 100 to yield 250 posterior samples per chain and so 1,000 posterior samples in total. We examined the convergence of Markov chain Monte Carlo by the potential scale-reduction factors[78] of the model parameters. We examined the explanatory power of the model through species-specific area under the curve[79] and Tjur's $R^2$ metric[80] values, which provide complementary insights of predictive performance[81].

To quantify the phylogenetic signals of climatic and seasonal variation we extracted four output variables for each species from the fitted HMSC models: climatic sensitivity, optimal climate, seasonal sensitivity and optimal season. We measured climatic sensitivity by the proportion of variance explained by the second-order polynomial of the MAT of the site. Similarly we measured seasonal sensitivity by the proportion of variance explained by the periodic functions $\sin(2\pi d/365)$ and $\cos(2\pi d/365)$. We multiplied the proportions of variance explained by the predictors out of the explained variation by the proportion of variation explained by the model, the latter measured by species-specific Tjur's $R^2$ values. We measured optimal climate as the MAT at which the second-order polynomial of the MAT was maximized, truncated to values within the observed range of MATs. Because it is meaningful to estimate the optimal climate only for species that show climatic variation, we included in the analyses of optimal climate only those species for which climatic sensitivity was at least 5%. Similarly we measured optimal season by the day of the year on which the estimated linear combination of the periodic functions $\sin(2\pi d/365)$ and $\cos(2\pi d/365)$ peaked, and included in the analyses of optimal season only those species for which seasonal sensitivity was at least 5%. We then fitted phylogenetic regression models for each of these four response variables and fitted the models with the R package nlme[73] using the gls function, no covariates and the corPagel correlation structure. We quantified the strength of the phylogenetic signal by the estimated λ parameter and estimated its statistical significance by the $P$ value of the comparison (performed by the analysis of variance function) between models that included versus did not include the corPagel correlation structure.

### Reporting summary

Further information on research design is available in the Nature Portfolio Reporting Summary linked to this article.

### Data availability

All data used in this paper are available at Zenodo (https://doi.org/10.5281/zenodo.10444737)[59]. GSSP data were downloaded from Ovaskainen et al.[3]. Climatic data were downloaded from the Copernicus Climate Change Service Climate Data Store[42] ('ERA5 hourly data on single levels dataset' and 'sis biodiversity era5 global dataset'). We extracted spore size and trophic guild data from data assembled by

Aguilar-Trigueros et al.[54]. Spore size data originate from species-level taxonomic descriptions in Mycobank[69].

## Code availability

The R pipeline that can be used to reproduce the results of this paper is available at Zenodo (https://doi.org/10.5281/zenodo.10444737)[59]. All analyses were conducted in R v.4.3.1 (ref. 82) with the packages ade4 1.7-22, adespatial 0.3-23, ape 5.7-1, ecmwfr 1.5.0, geosphere 1.5-18, Hmsc 3.0-14, jsonify 1.2.2, kgc 1.0.0.2, lme4 1.1-35.1, lubridate 1.9.3, maps 3.4.2, MASS 7.3-60, ncdf4 1.22, nlme 3.1-162, phyloseq 1.46.0, phytools 2.1-1, raster 3.6-26, rgdal 1.6-7, scales 1.3.0, sjstats 0.18.2, tidyverse 2.0.0, vegan 2.6.4, VennDiagram 1.7.3, vioplot 0.4.0 and wordcloud 2.6.

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

**Acknowledgements** We acknowledge H. Aho, J. Frietsch, T. Kankaanpää, J. Koskinen, B. McDonald, T. McDermott, E. Meyke, M. Mjomba, P. A. Niklaus, G. Saint-Jean, M. Tiusanen, H. Wirta, V. Zengerer and several UCSC students for their contributions in data sampling and for technical assistance. This study was supported by funding from the Academy of Finland (grant nos. 336212, 345110, 322266, 335354 and 357475); the European Research Council (ERC) under the European Union's Horizon 2020 research and innovation programme (grant agreement no. 856506; ERC-synergy project LIFEPLAN); the EU Horizon 2020 project INTERACT under grant agreement nos. 730938 and 871120; the Jane and Aatos Erkko Foundation; the Research Council of Norway through its Centres of Excellence Funding Scheme (no. 223257); the Estonian Research Council (grant no. PRG1170); FORMAS (grant nos. 215-2011-498 and 226-2014-1109); the Canada Foundation for Innovation, Polar Knowledge Canada, Natural Sciences and Engineering Research Council of Canada (NSERC Discover); Natural Environment Research Council (NERC) UK (grant nos. NE/N001710/1 and NE/N002431/1); BBSRC (grant no. BB/L012286/1); the Austrian Ministry of Science (the ABOL-HRSM project); the municipality of Vienna (Division of Environmental Protection); Southern Scientific Centre RAS (project no. 122020100332-8); the Croatian Science Foundation under the project FunMed (grant no. HRZZ-IP-2022-10-5219); the National Research Council of Thailand (grant no. N42A650547); Dirigibile Italia Station, Institute of Polar Science (ISP) – National Research Council; the US National Science Foundation (nos. DEB-1655896, DEB-1655076 and DEB-1932467); the Pepper-Giberson Chair Fund; the National Science Foundation of China (grant nos. 41761144055 and 41771063); São Paulo Research Foundation (no. FAPESP 2016/25197-0) and Legado das Águas-Brazil; Hong Kong's Research Grants Council (General Research Fund no. 17118317); the Norwegian Institute for Nature Research; Canada's New Frontiers in Research Fund; Swedish Research Council support (grant no. 4.3-2021-00164) to SITES and Abisko Scientific Research Station; the Mushroom Research Foundation, Thailand; and the Italian National Biodiversity Future Center (MUR-PNRR, Mission 4.2. Investment 1.4, Project no. CN00000033).

**Author contributions** N.A. conceived the study, led the analyses and wrote the first draft of the manuscript. B.F. led the development of the bioinformatics pipeline and contributed to the first draft of the manuscript. B.H. participated in project coordination, participated in sample preparation and commented on the manuscript. P.S. contributed to the development of the bioinformatics pipeline and commented on the manuscript. I.P. acquired data, participated in project coordination, participated in sample preparation and commented on the manuscript. C.A.A.-T. contributed to trait-based analyses and commented on the manuscript. N.R.A. acquired data and commented on the manuscript. U.V.B. acquired data and commented on the manuscript. T.B. acquired data and commented on the manuscript. G.B. acquired data and commented on the manuscript. S.N.B. acquired data and commented on the manuscript. T.C.B. acquired data and commented on the manuscript. G.L.B. acquired data and commented on the manuscript. S.B.-H. acquired data and commented on the manuscript. C.B. acquired data and commented on the manuscript. L.C. acquired data and commented on the manuscript. E.K.C. acquired data and commented on the manuscript. E.C. participated in sample preparation and commented on the manuscript. S.C. acquired data and commented on the manuscript. L.P.D. acquired data and commented on the manuscript. N.d.V. acquired data and commented on the manuscript. M.-L.D.-L. acquired data and commented on the manuscript. M.A.K.D. acquired data and commented on the manuscript. I.B.D.J. acquired data and commented on the manuscript. B.L.F. acquired data and commented on the manuscript. M.f.d.J. acquired data and commented on the manuscript. G.S.G. acquired data and commented on the manuscript. G.W.G. acquired data and commented on the manuscript. A.A.G. acquired data and commented on the manuscript. A.G. acquired data and commented on the manuscript. H.G. acquired data and commented on the manuscript. P.H. contributed to data processing and commented on the manuscript. R.H. acquired data and commented on the manuscript. J. Hansen acquired data and commented on the manuscript. L.H.H. acquired data and commented on the manuscript. A.D.M.T.H. acquired data and commented on the manuscript. S.H. acquired data and commented on the manuscript. I.D.H. acquired data and commented on the manuscript. J. Hultman contributed to the development of the bioinformatics pipeline and commented on the manuscript. K.D.H. acquired data and commented on the manuscript. N.A.H. acquired data and commented on the manuscript. N.I. contributed to the planning and implementation of DNA extraction and sequencing and commented on the manuscript. P.K. acquired data and commented on the manuscript. D.K. participated in project coordination, participated in sample preparation and commented on the manuscript. A.K. acquired data and commented on the manuscript. I.K.-G. acquired data and commented on the manuscript. J.K. facilitated data acquirement and commented on the manuscript. M.K. contributed to the planning and implementation of DNA extraction and sequencing and commented on the manuscript. N.L. acquired data and commented on the manuscript. E. Lecomte acquired data and commented on the manuscript. V.L. acquired data and commented on the manuscript. E. Lundin acquired data and commented on the manuscript. A. Meire acquired data and commented on the manuscript. A. Mešić acquired data and commented on the manuscript. O.M. contributed to data processing and commented on the manuscript. N.M. contributed to the planning and implementation of DNA extraction and sequencing and commented on the manuscript. P.M. acquired data and commented on the manuscript. J.M. acquired data and commented on the manuscript. R.H.N. facilitated data acquirement and commented on the manuscript. P.Y.C.N. acquired data and commented on the manuscript. J.N. acquired data and commented on the manuscript. B.N. acquired data and commented on the manuscript. V.N. participated in data interpretation and commented on the manuscript. C. Paz acquired data and commented on the manuscript. P.P. acquired data and commented on the manuscript. D.P. acquired data and commented on the manuscript. G.P. acquired data and commented on the manuscript. J.-M.P. participated in project coordination, participated in sample preparation and commented on the manuscript. F.P. acquired data and commented on the manuscript. C. Potter acquired data and commented on the manuscript. J.P. contributed to data processing and commented on the manuscript. S.P. acquired data and commented on the manuscript. A. Rafiq acquired data and commented on the manuscript. D.R. acquired data and commented on the manuscript. N.R. acquired data and commented on the manuscript. A.R.R. acquired data and commented on the manuscript. K. Raundrup acquired data and commented on the manuscript. Y.A.R. acquired data and commented on the manuscript. J.R. acquired data and commented on the manuscript. H.M.R. participated in project coordination, participated in sample preparation and commented on the manuscript. A. Rogovsky acquired data and commented on the manuscript. Y.R. acquired data and commented on the manuscript. K. Runnel acquired data and commented on the manuscript. A. Saarto acquired data and commented on the manuscript. A. Savchenko contributed to data processing and commented on the manuscript. M.S. acquired data and commented on the manuscript. N.M.S. acquired data and commented on the manuscript. S.S. acquired data and commented on the manuscript. C.S. acquired data and commented on the manuscript. E.S. acquired data and commented on the manuscript. S.V.S. acquired data and commented on the manuscript. I.S. acquired data and commented on the manuscript. L. Tedersoo acquired data and commented on the manuscript. J.T. acquired data and commented on the manuscript. L. Tipton acquired data and commented on the manuscript. H.T. acquired data and commented on the manuscript. M.U.-P. participated in sample preparation and commented on the manuscript. M.v.d.B. acquired data and commented on the manuscript. F.H.v.d.B. acquired data and commented on the manuscript. B.V. acquired data and commented on the manuscript. S.V. acquired data and commented on the manuscript. S.R.V. acquired data and commented on the manuscript. X.W. acquired data and commented on the manuscript. W.W.W. acquired data and commented on the manuscript. S.N.W. acquired data and commented on the manuscript. S.J.W. acquired data and commented on the manuscript. C.Y. acquired data and commented on the manuscript. N.S.Y. acquired data and commented on the manuscript. A.Y. acquired data and commented on the manuscript. D.W.Y. acquired data and commented on

on the manuscript. E.V.Z. contributed to the planning and implementation of DNA extraction and sequencing and commented on the manuscript. P.D.N.H. contributed to the planning and implementation of DNA extraction and sequencing and commented on the manuscript. T.R. conceived the study and contributed to the first draft of the manuscript. O.O. conceived the study, contributed to analyses and contributed to the first draft of the manuscript.

**Competing interests** The authors declare no competing interests.

**Additional information**
**Correspondence and requests for materials** should be addressed to Nerea Abrego.

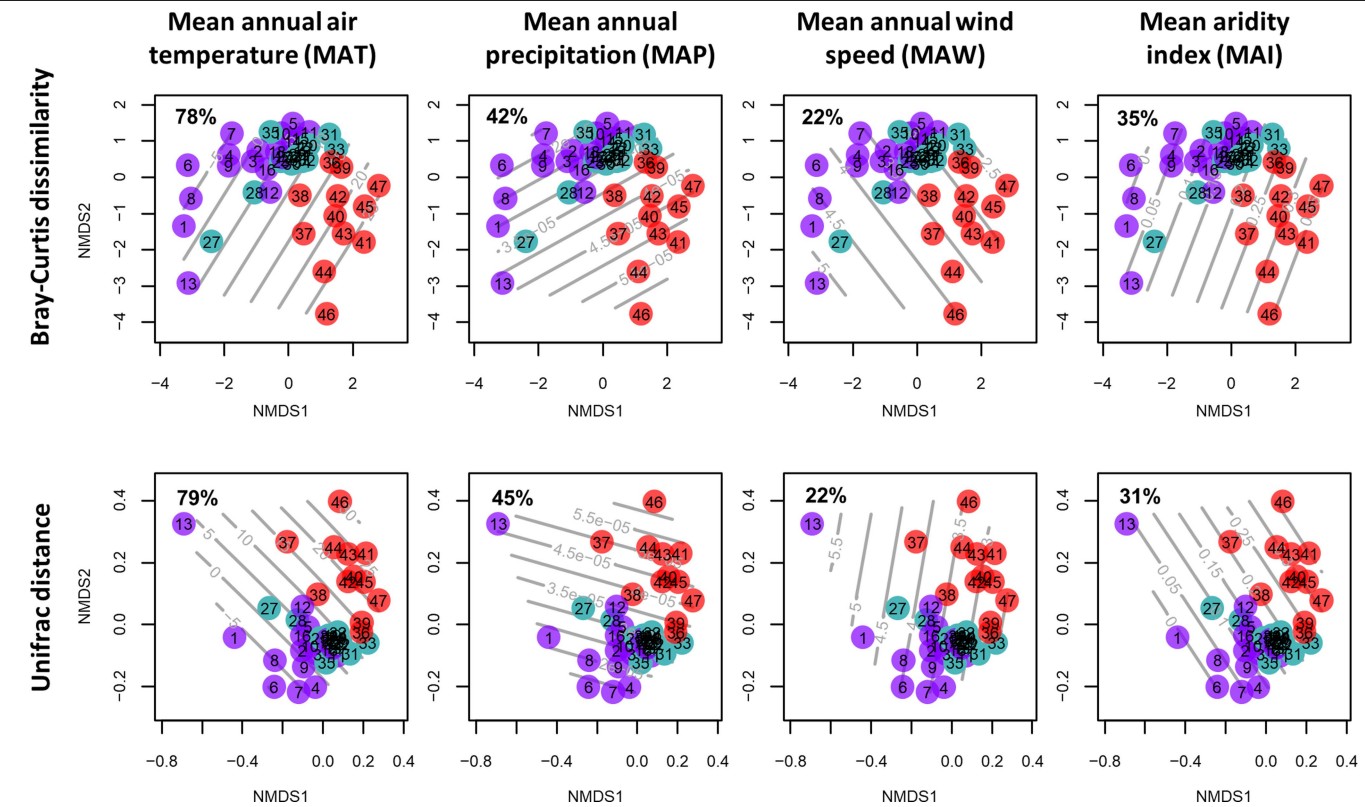

**Extended Data Fig. 1 | Results of ordination analyses.** The upper panels show the results for the Bray-Curtis dissimilarity, and the lower panels for the Unifrac distance. The bolded percentages show the proportion of deviance in the ordination space explained by the focal environmental covariate: MAT (1st column of panels), MAP (2nd column of panels), MAW (3rd column of panels), or MAI (4th column of panels).

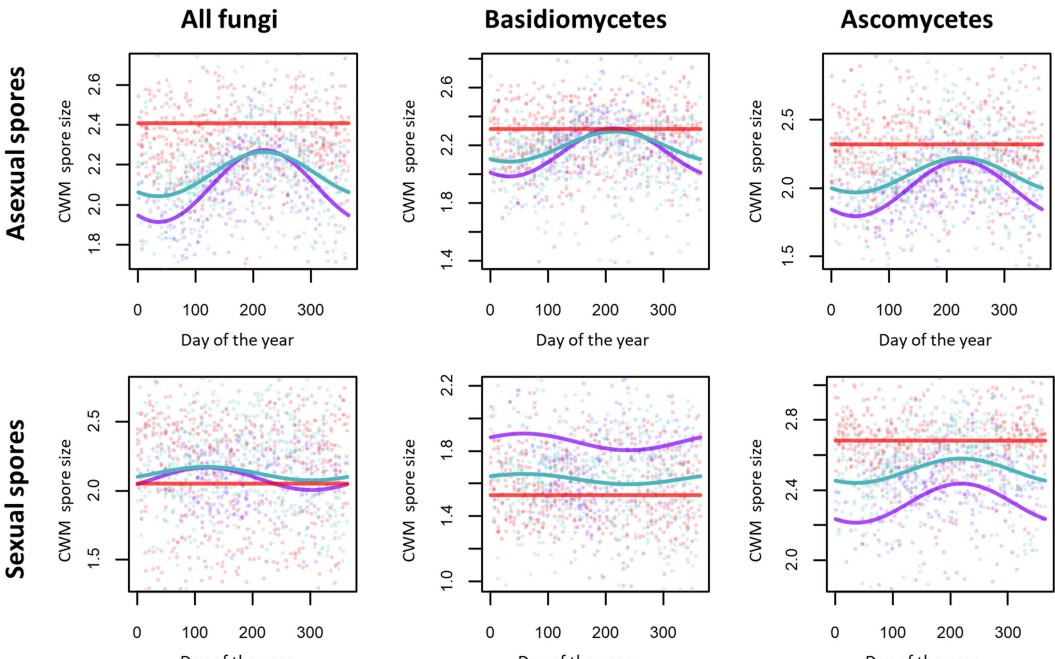

**Extended Data Fig. 2 | Seasonal variation in community weighted mean spore size.** The lines in the panels show the predictions of the best-supported linear mixed models (see Methods) for tropical-subtropical (red), temperate (cyan), and polar-continental (purple) climatic zones. Note that the predictions are shown for the Northern Hemisphere, whereas for the Southern Hemisphere the seasonal patterns would be mirror images. The response variable in each panel is the community weighted mean spore size for the type of spores shown in the row and column titles. The dots that show the raw data have been slightly jittered to reveal overlap.

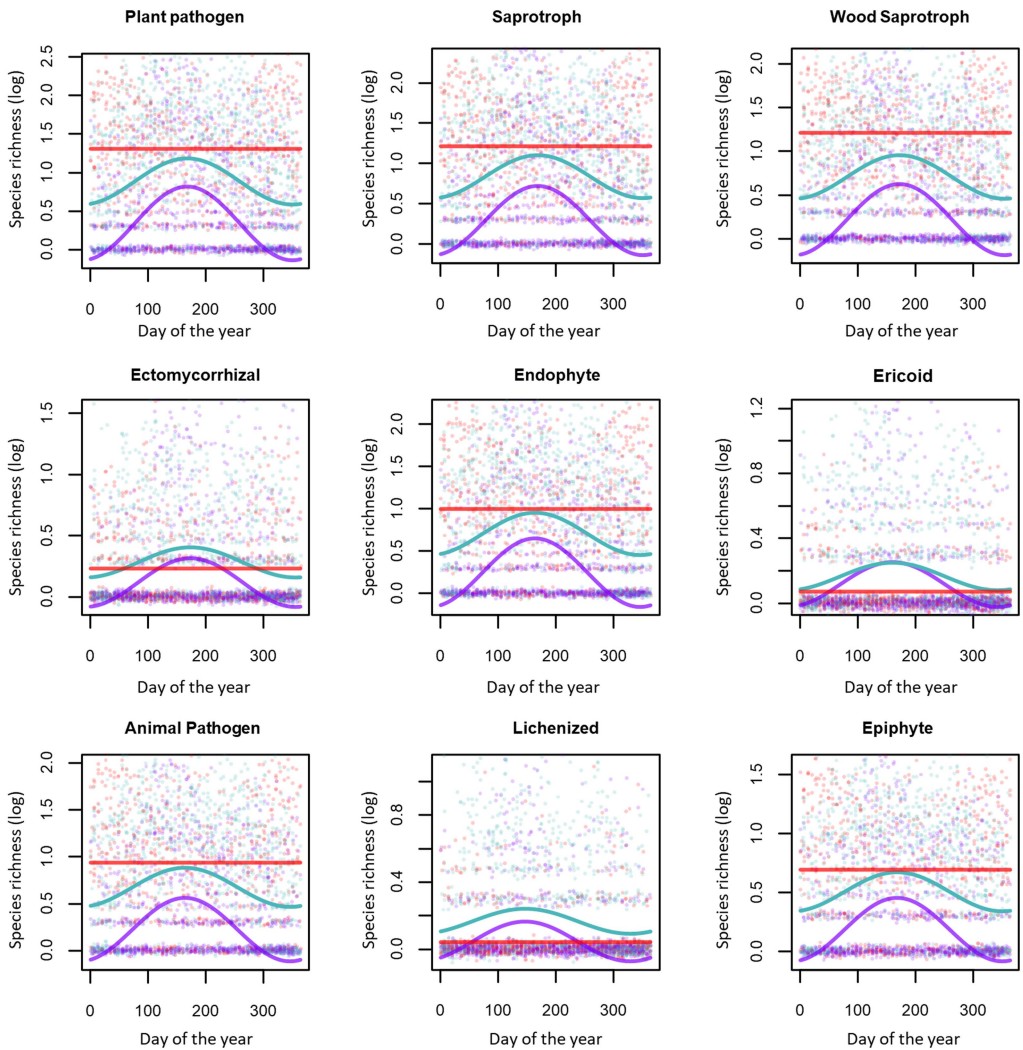

**Extended Data Fig. 3 | Seasonal variation in airborne fungal diversity of different trophic guilds.** The lines in the panels show the predictions of the best-supported linear mixed models (see Methods) for tropical-subtropical (red), temperate (cyan), and polar-continental (purple) climatic zones. Note that the predictions are shown for the Northern Hemisphere, whereas for the Southern Hemisphere the seasonal patterns would be mirror images. The response variable in each panel is the species richness of the trophic guild shown indicated on top of the panels. The dots that show the raw data have been slightly jittered to reveal overlap.

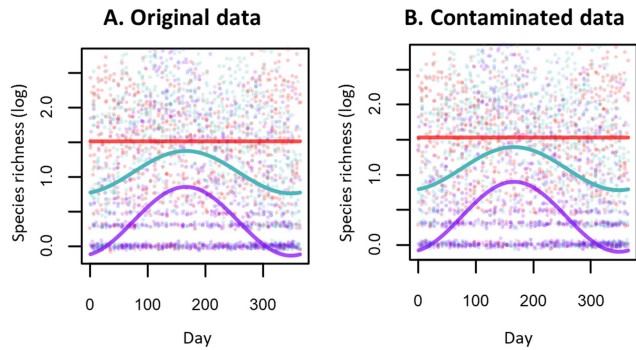

**Extended Data Fig. 4 | Comparison species richness analyses with original data (A; replicated from Fig. 3a of the main document) and data contaminated with sequences counts observed in negative controls (B).** The results are shown for one of the ten replicates of the contaminated datasets.

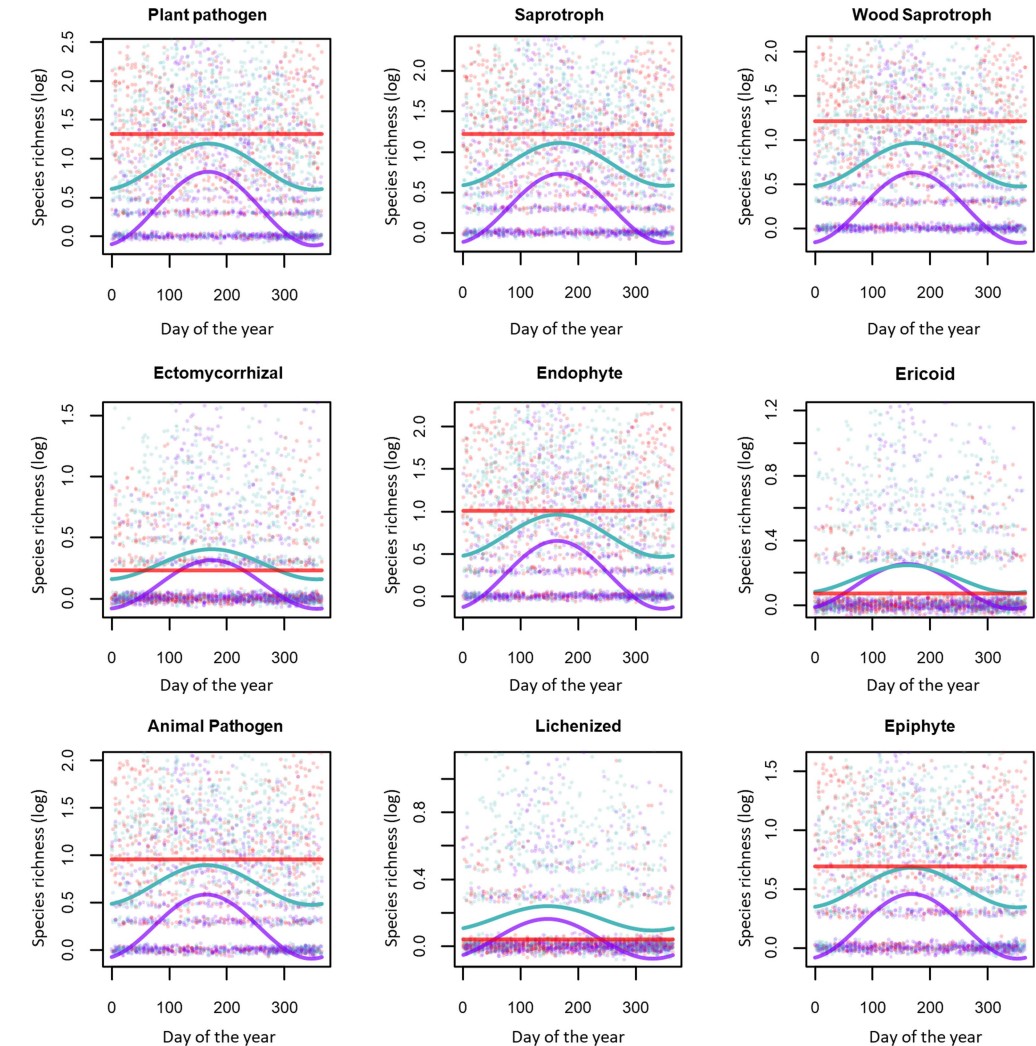

**Extended Data Fig. 5 | Seasonal variation in airborne fungal diversity of different trophic guilds, with data contaminated with sequences counts observed in negative controls.** The lines in the panels show the predictions of the best-supported linear mixed models (see Methods) for tropical-subtropical (red), temperate (cyan), and polar-continental (purple) climatic zones. Note that the predictions are shown for the Northern Hemisphere, whereas for the Southern Hemisphere the seasonal patterns would be mirror images. The response variable in each panel is the species richness of the trophic guild shown in the panel's titles. The dots in that show the raw data have been slightly jiggered to reveal overlap. For comparison with the original data, see Extended Data Fig. 5. The results are shown for one of the ten replicates of the contaminated datasets.

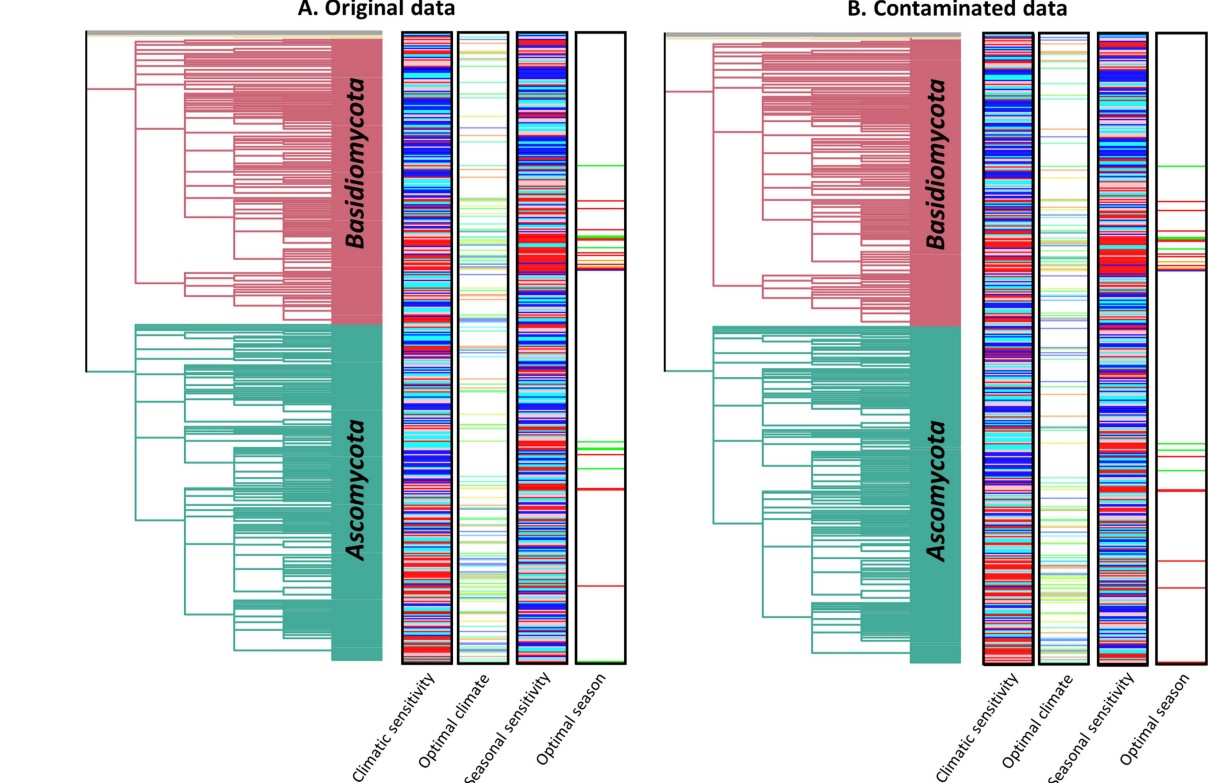

**A. Original data**

**B. Contaminated data**

*Basidiomycota*

*Ascomycota*

Climatic sensitivity

Optimal climate

Seasonal sensitivity

Optimal season

**Extended Data Fig. 6 | Comparison of HMSC analyses with original data (A; replicated from Fig. 4a of the main document) and data contaminated with sequences counts observed in negative controls (B).** The results are shown for one of the ten replicates of the contaminated datasets.

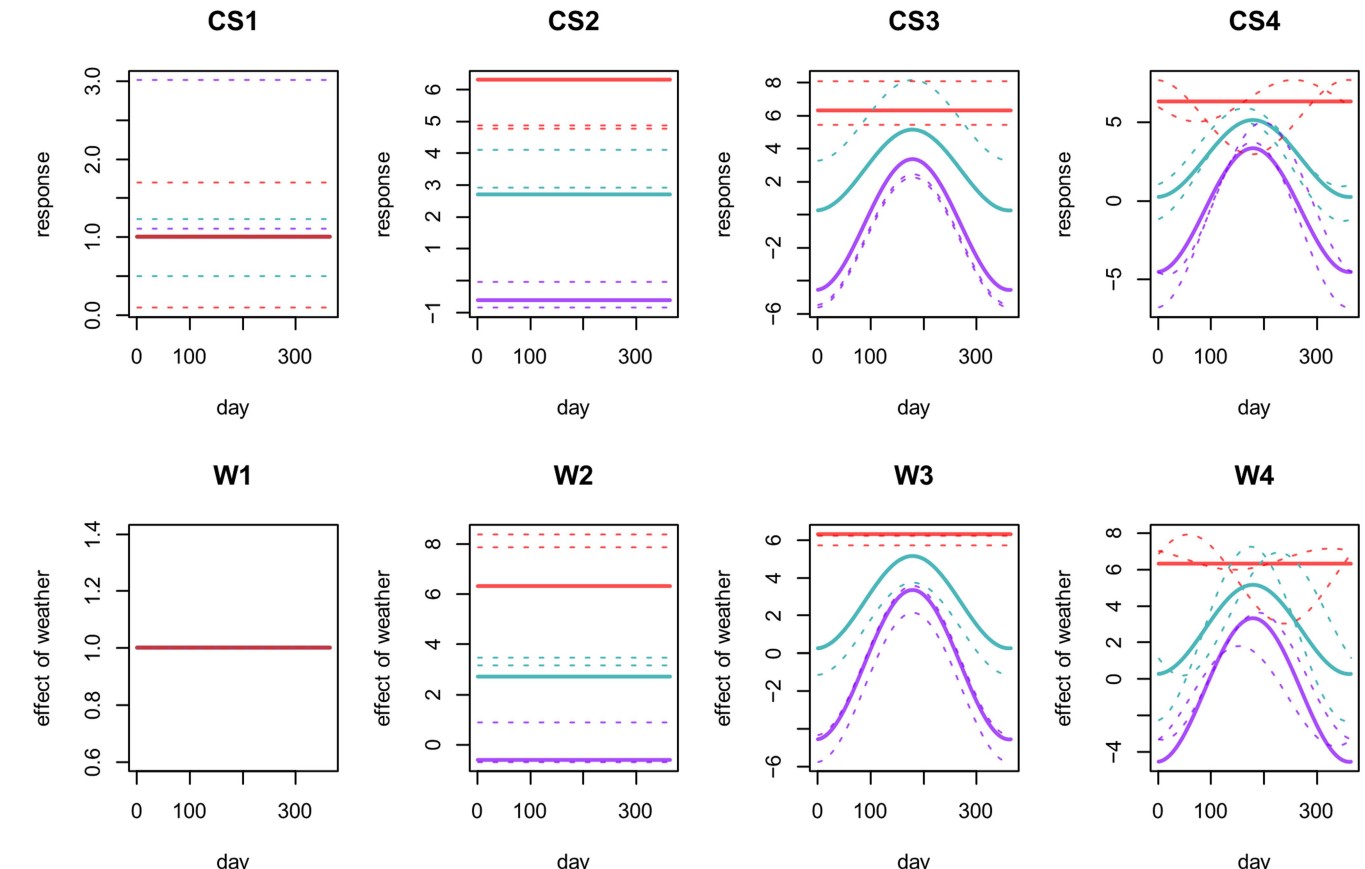

**Extended Data Fig. 7 | A conceptual illustration of different model variants.** The upper panels depict the climatic models CS1 (Null; site-dependency as random effect), CS2 (Climate-dependency), CS3 (Climate-dependency and latitude-dependent seasonality) and CS4 (Climate-dependency and site-specific seasonality). The lower panels depict the weather models W1 (Constant weather effects), W2 (Weather effects depend on site), W3 (Weather effects depend on site and on latitude-dependent seasonality), and W4 (Weather effects depend on site and on site-dependent seasonality). Note that the weather models are not independent models as they are embedded within the climatic models. In each panel, the lines show hypothetical responses. The continuous lines depict the mean responses for the tropical-subtropical (red), temperate (cyan), and polar-continental (purple) climatic zones.

**Extended Data Table 1 | Most common genera found in the GSSP data**

| Climatic zone | | | All | | Polar-continental | | Temperate | | Tropical-subtropical | |
|---|---|---|---|---|---|---|---|---|---|---|
| Phylum | Order | Genus | % | Rank | % | Rank | % | Rank | % | Rank |
| Ascomycota | Capnodiales | Cladosporium | 53 | 1 | 36 | 1 | 60 | 1 | 68 | 1 |
| Ascomycota | Pleosporales | Ascochyta | 40 | 2 | 19 | 3 | 50 | 2 | 52 | 2 |
| Ascomycota | Pleosporales | Alternaria | 34 | 3 | 18 | 4 | 43 | 3 | 40 | 4 |
| Basidiomycota | Tremellales | Cryptococcus | 33 | 4 | 22 | 2 | 39 | 4 | 38 | 6 |
| Ascomycota | Dothideales | Aureobasidium | 28 | 5 | 12 | 10 | 38 | 5 | 35 | 9 |
| Basidiomycota | Polyporales | Trametes | 28 | 6 | 12 | 11 | 35 | 6 | 38 | 5 |
| Ascomycota | Pleosporales | Phaeosphaeria | 25 | 7 | 15 | 7 | 32 | 8 | 27 | 13 |
| Ascomycota | Pleosporales | Phoma | 22 | 8 | 09 | 28 | 26 | 13 | 36 | 7 |
| Basidiomycota | Tremellales | Dioszegia | 22 | 9 | 16 | 6 | 32 | 7 | 15 | 48 |
| Ascomycota | Chaetothyriales | Capronia | 22 | 10 | 13 | 8 | 29 | 9 | 24 | 18 |
| Basidiomycota | Cantharellales | Sistotrema | 21 | 11 | 16 | 5 | 25 | 14 | 19 | 29 |
| Ascomycota | Helotiales | pseudogenus | 19 | 16 | 13 | 9 | 28 | 12 | 13 | 65 |
| Basidiomycota | Filobasidiales | Filobasidium | 19 | 17 | 11 | 16 | 29 | 10 | 14 | 59 |
| Ascomycota | Pleosporales | Periconia | 20 | 14 | 07 | 49 | 18 | 32 | 41 | 3 |
| Basidiomycota | Polyporales | Phlebia | 20 | 12 | 10 | 21 | 21 | 20 | 36 | 8 |
| Ascomycota | Capnodiales | Pseudocercospora | 14 | 25 | 04 | 121 | 12 | 74 | 33 | 10 |

The table shows the prevalence (%) of each genus, computed as the fraction of samples in which it was detected, as well as the ranking of the genus in terms of its prevalence. The prevalences and rankings are shown for all samples, as well as separately for the samples from each of the three climatic zones. Genera that rank in the top ten are highlighted, and only genera that ranked in the top ten in at least one of the climatic zones are included.

**Extended Data Table 2 | The numbers of OTUs classified into trophic guilds used in this study**

| Trophic guild | #OTUs |
|---|---|
| Plant pathogen | 12539 |
| Saprotroph (including dung saprotroph) | 10380 |
| Wood saprotroph | 4856 |
| Endophyte | 7480 |
| Ericoid mycorrhizal | 610 |
| Ectomycorrhizal | 1698 |
| Animal pathogen (including human pathogens) | 4109 |
| Lichenized | 494 |
| Epiphyte | 1316 |

Note that each OTU may be classified to more than one trophic guild and hence the sum of #OTUs over the trophic guilds exceeds the total number of OTUs detected in our study.

# Reporting Summary

## Statistics

For all statistical analyses, confirm that the following items are present in the figure legend, table legend, main text, or Methods section.

| n/a | Confirmed | |
|---|---|---|
| ☐ | ☒ | The exact sample size (*n*) for each experimental group/condition, given as a discrete number and unit of measurement |
| ☐ | ☒ | A statement on whether measurements were taken from distinct samples or whether the same sample was measured repeatedly |
| ☐ | ☒ | The statistical test(s) used AND whether they are one- or two-sided<br>*Only common tests should be described solely by name; describe more complex techniques in the Methods section.* |
| ☐ | ☒ | A description of all covariates tested |
| ☐ | ☒ | A description of any assumptions or corrections, such as tests of normality and adjustment for multiple comparisons |
| ☐ | ☒ | A full description of the statistical parameters including central tendency (e.g. means) or other basic estimates (e.g. regression coefficient) AND variation (e.g. standard deviation) or associated estimates of uncertainty (e.g. confidence intervals) |
| ☒ | ☐ | For null hypothesis testing, the test statistic (e.g. *F*, *t*, *r*) with confidence intervals, effect sizes, degrees of freedom and *P* value noted<br>*Give P values as exact values whenever suitable.* |
| ☐ | ☒ | For Bayesian analysis, information on the choice of priors and Markov chain Monte Carlo settings |
| ☒ | ☐ | For hierarchical and complex designs, identification of the appropriate level for tests and full reporting of outcomes |
| ☐ | ☒ | Estimates of effect sizes (e.g. Cohen's *d*, Pearson's *r*), indicating how they were calculated |

*Our web collection on statistics for biologists contains articles on many of the points above.*

## Software and code

Policy information about availability of computer code

| Data collection | Software was not used for data collection |
|---|---|
| Data analysis | Custom codes were central to the conclusions of the paper. The R-pipeline that can be used to reproduce the results of this paper are available at Zenodo (Abrego et al. 2024. Data and scripts for: Airborne DNA reveals predictable spatial and seasonal dynamics of fungi. https://doi.org/10.5281/zenodo.10444737). All analyses were conducted with R version 4.3.1 (R Core Team. 2023. R: A language and environment for statistical computing. R Foundation for Statistical Computing), with the R-packages ade4 1.7-22, adespatial 0.3-23, ape 5.7-1, ecmwfr 1.5.0, geosphere 1.5-18, Hmsc 3.0-14, jsonify 1.2.2, kgc 1.0.0.2, lme4 1.1-35.1, lubridate 1.9.3, maps 3.4.2, MASS 7.3-60, ncdf4 1.22, nlme 3.1-162, phyloseq 1.46.0, phytools 2.1-1, raster 3.6-26, rgdal 1.6-7, scales 1.3.0, sjstats 0.18.2, tidyverse 2.0.0, vegan 2.6.4, VennDiagram 1.7.3, vioplot 0.4.0 and wordcloud 2.6. |

For manuscripts utilizing custom algorithms or software that are central to the research but not yet described in published literature, software must be made available to editors and reviewers. We strongly encourage code deposition in a community repository (e.g. GitHub). See the Nature Portfolio guidelines for submitting code & software for further information.

# Data

Policy information about availability of data

All manuscripts must include a data availability statement. This statement should provide the following information, where applicable:
- Accession codes, unique identifiers, or web links for publicly available datasets
- A description of any restrictions on data availability
- For clinical datasets or third party data, please ensure that the statement adheres to our policy

All data used in this paper are available at Zenodo (Abrego et al. 2024. Data and scripts for: Airborne DNA reveals predictable spatial and seasonal dynamics of fungi. https://doi.org/10.5281/zenodo.10444737). The GSSP data were downloaded from Ovaskainen et al. (Ovaskainen, O. et al. Global Spore Sampling Project: A global, standardized dataset of airborne fungal DNA. Revision under consideration in Scientific Data). The climatic data were downloaded from the Copernicus Climate Change Service (C3S) Climate Data Store ("ERA5 hourly data on single levels dataset" and "sis biodiversity era5 global dataset"; Hersbach, H. et al. 2020. The ERA5 global reanalysis. Quart J Royal Meteoro Soc 146, 1999–2049). We extracted the spore size and trophic guild data from the data assembled by Aguilar-Trigueros et al54. The spore size data originates from species-level taxonomic descriptions in Mycobank (Robert, V. et al. 2013. MycoBank gearing up for new horizons. IMA Fungus 4, 371–379).

# Research involving human participants, their data, or biological material

Policy information about studies with human participants or human data. See also policy information about sex, gender (identity/presentation), and sexual orientation and race, ethnicity and racism.

| | |
|---|---|
| Reporting on sex and gender | The research does not contain human participants or human data. |
| Reporting on race, ethnicity, or other socially relevant groupings | The research does not contain human participants or human data. |
| Population characteristics | The research does not contain human participants or human data. |
| Recruitment | The research does not contain human participants or human data. |
| Ethics oversight | The research does not contain human participants or human data. |

Note that full information on the approval of the study protocol must also be provided in the manuscript.

# Field-specific reporting

Please select the one below that is the best fit for your research. If you are not sure, read the appropriate sections before making your selection.

☐ Life sciences        ☐ Behavioural & social sciences        ☒ Ecological, evolutionary & environmental sciences

For a reference copy of the document with all sections, see nature.com/documents/nr-reporting-summary-flat.pdf

# Life sciences study design

All studies must disclose on these points even when the disclosure is negative.

| | |
|---|---|
| Sample size | Describe how sample size was determined, detailing any statistical methods used to predetermine sample size OR if no sample-size calculation was performed, describe how sample sizes were chosen and provide a rationale for why these sample sizes are sufficient. |
| Data exclusions | Describe any data exclusions. If no data were excluded from the analyses, state so OR if data were excluded, describe the exclusions and the rationale behind them, indicating whether exclusion criteria were pre-established. |
| Replication | Describe the measures taken to verify the reproducibility of the experimental findings. If all attempts at replication were successful, confirm this OR if there are any findings that were not replicated or cannot be reproduced, note this and describe why. |
| Randomization | Describe how samples/organisms/participants were allocated into experimental groups. If allocation was not random, describe how covariates were controlled OR if this is not relevant to your study, explain why. |
| Blinding | Describe whether the investigators were blinded to group allocation during data collection and/or analysis. If blinding was not possible, describe why OR explain why blinding was not relevant to your study. |

# Behavioural & social sciences study design

All studies must disclose on these points even when the disclosure is negative.

| | |
|---|---|
| Study description | *Briefly describe the study type including whether data are quantitative, qualitative, or mixed-methods (e.g. qualitative cross-sectional, quantitative experimental, mixed-methods case study).* |
| Research sample | *State the research sample (e.g. Harvard university undergraduates, villagers in rural India) and provide relevant demographic information (e.g. age, sex) and indicate whether the sample is representative. Provide a rationale for the study sample chosen. For studies involving existing datasets, please describe the dataset and source.* |
| Sampling strategy | *Describe the sampling procedure (e.g. random, snowball, stratified, convenience). Describe the statistical methods that were used to predetermine sample size OR if no sample-size calculation was performed, describe how sample sizes were chosen and provide a rationale for why these sample sizes are sufficient. For qualitative data, please indicate whether data saturation was considered, and what criteria were used to decide that no further sampling was needed.* |
| Data collection | *Provide details about the data collection procedure, including the instruments or devices used to record the data (e.g. pen and paper, computer, eye tracker, video or audio equipment) whether anyone was present besides the participant(s) and the researcher, and whether the researcher was blind to experimental condition and/or the study hypothesis during data collection.* |
| Timing | *Indicate the start and stop dates of data collection. If there is a gap between collection periods, state the dates for each sample cohort.* |
| Data exclusions | *If no data were excluded from the analyses, state so OR if data were excluded, provide the exact number of exclusions and the rationale behind them, indicating whether exclusion criteria were pre-established.* |
| Non-participation | *State how many participants dropped out/declined participation and the reason(s) given OR provide response rate OR state that no participants dropped out/declined participation.* |
| Randomization | *If participants were not allocated into experimental groups, state so OR describe how participants were allocated to groups, and if allocation was not random, describe how covariates were controlled.* |

# Ecological, evolutionary & environmental sciences study design

All studies must disclose on these points even when the disclosure is negative.

| | |
|---|---|
| Study description | The data were collected by the Global Spore Sampling Project (GSSP). The GSSP involves 47 sampling locations distributed across all continents except Antarctica, with each location collecting two 24-hr samples per week, in most cases over a period of one year or more. Sampling is conducted with a cyclone sampler, which orients itself in the direction of the wind and collects particles >1 μm in size from the air directly into a sampling tube with a single reverse-flow cyclone. For DNA sequencing, we targeted part of the nuclear ribosomal internal transcribed spacer (ITS) region, which is the universal molecular barcode for fungi. |
| Research sample | A research sample is one sampling vial, to which we collected by a cyclone sampler with 24-hr period the particles from the air. |
| Sampling strategy | We targeted to obtain uninterrupted weekly time-series data for at least one year (to cover the full season) from as many and as globally representative locations as possible. |
| Data collection | The authors collected the data by applying the cyclone sampler, as explained in the separate data paper. |
| Timing and spatial scale | As shown in Fig. 1AB of the manuscript, the sampling covers all continents except Antarctica, and it is most intensive in Europe. The data from each site is a weekly time series, starting earliest in Autumn 2019 and continuing until beginning of 2021. There are some gaps in the weekly sampling interval due to logistic constraints. |
| Data exclusions | No data were excluded in the analysis (except for some specific analysis, as mentioned in the methods). |
| Reproducibility | We have taken extra samples so that the reproducibility of empirical data can be addressed. These additional samples are stored but not sequenced. All analyses can be reproduced by the codes provided in Zenodo (Abrego et al. 2024. Data and scripts for: Airborne DNA reveals predictable spatial and seasonal dynamics of fungi. https://doi.org/10.5281/zenodo.10444737) |
| Randomization | This is not relevant for the study because we targeted for regular (=weekly) samples to study seasonality. The globally distributed sampling locations were not randomized as they were based on the research groups who expressed their willingness to take part to the study (who are co-authors of the paper). |
| Blinding | Blinding was not applicable for this study, as all analyses were conducted algorithmically with automated pipelines (provided in Zenodo). |

Did the study involve field work? 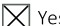 ☒ Yes  ☐ No

## Field work, collection and transport

| | |
|---|---|
| Field conditions | The study involves 47 globally distributed sites. Their climatic conditions, as well as the weather conditions at the time of sampling, are provided in the metadata. |
| Location | Global. |
| Access & import/export | The methodology is based on air sampling and is thus completely non-invasive. All participating teams were responsible for obtaining any relevant permits, such as a permit from the land-owner to place the cyclone sampler to the site over the sampling period (e.g., for the sampler placed at University of Helsinki, we obtained permit from the Viikki campus).  Concerning the export/import of the samples from the sampling locations to Canada, we searched but failed to find any legislation that would require permits to sample air. Thus, while there are fairly strict rules on importing  live animals/plants and soil,  and on some food products, such as  meat or dairy products, we could not find restrictions on air samples. |
| Disturbance | The methodology is based on air sampling and is thus completely non-invasive. Disturbance was thus restricted to researchers visiting the sites. |

# Reporting for specific materials, systems and methods

We require information from authors about some types of materials, experimental systems and methods used in many studies. Here, indicate whether each material, system or method listed is relevant to your study. If you are not sure if a list item applies to your research, read the appropriate section before selecting a response.

### Materials & experimental systems

| n/a | Involved in the study |
|---|---|
| ☒ | ☐ Antibodies |
| ☒ | ☐ Eukaryotic cell lines |
| ☒ | ☐ Palaeontology and archaeology |
| ☒ | ☐ Animals and other organisms |
| ☒ | ☐ Clinical data |
| ☒ | ☐ Dual use research of concern |
| ☒ | ☐ Plants |

### Methods

| n/a | Involved in the study |
|---|---|
| ☒ | ☐ ChIP-seq |
| ☒ | ☐ Flow cytometry |
| ☒ | ☐ MRI-based neuroimaging |

## Antibodies

| | |
|---|---|
| Antibodies used | *Describe all antibodies used in the study; as applicable, provide supplier name, catalog number, clone name, and lot number.* |
| Validation | *Describe the validation of each primary antibody for the species and application, noting any validation statements on the manufacturer's website, relevant citations, antibody profiles in online databases, or data provided in the manuscript.* |

## Eukaryotic cell lines

Policy information about cell lines and Sex and Gender in Research

| | |
|---|---|
| Cell line source(s) | *State the source of each cell line used and the sex of all primary cell lines and cells derived from human participants or vertebrate models.* |
| Authentication | *Describe the authentication procedures for each cell line used OR declare that none of the cell lines used were authenticated.* |
| Mycoplasma contamination | *Confirm that all cell lines tested negative for mycoplasma contamination OR describe the results of the testing for mycoplasma contamination OR declare that the cell lines were not tested for mycoplasma contamination.* |
| Commonly misidentified lines (See ICLAC register) | *Name any commonly misidentified cell lines used in the study and provide a rationale for their use.* |

## Palaeontology and Archaeology

| | |
|---|---|
| Specimen provenance | *Provide provenance information for specimens and describe permits that were obtained for the work (including the name of the issuing authority, the date of issue, and any identifying information). Permits should encompass collection and, where applicable, export.* |
| Specimen deposition | *Indicate where the specimens have been deposited to permit free access by other researchers.* |

| Dating methods | *If new dates are provided, describe how they were obtained (e.g. collection, storage, sample pretreatment and measurement), where they were obtained (i.e. lab name), the calibration program and the protocol for quality assurance OR state that no new dates are provided.* |

☐ Tick this box to confirm that the raw and calibrated dates are available in the paper or in Supplementary Information.

| Ethics oversight | *Identify the organization(s) that approved or provided guidance on the study protocol, OR state that no ethical approval or guidance was required and explain why not.* |

Note that full information on the approval of the study protocol must also be provided in the manuscript.

# Animals and other research organisms

Policy information about studies involving animals; ARRIVE guidelines recommended for reporting animal research, and Sex and Gender in Research

| Laboratory animals | *For laboratory animals, report species, strain and age OR state that the study did not involve laboratory animals.* |
| Wild animals | *Provide details on animals observed in or captured in the field; report species and age where possible. Describe how animals were caught and transported and what happened to captive animals after the study (if killed, explain why and describe method; if released, say where and when) OR state that the study did not involve wild animals.* |
| Reporting on sex | *Indicate if findings apply to only one sex; describe whether sex was considered in study design, methods used for assigning sex. Provide data disaggregated for sex where this information has been collected in the source data as appropriate; provide overall numbers in this Reporting Summary. Please state if this information has not been collected. Report sex-based analyses where performed, justify reasons for lack of sex-based analysis.* |
| Field-collected samples | *For laboratory work with field-collected samples, describe all relevant parameters such as housing, maintenance, temperature, photoperiod and end-of-experiment protocol OR state that the study did not involve samples collected from the field.* |
| Ethics oversight | *Identify the organization(s) that approved or provided guidance on the study protocol, OR state that no ethical approval or guidance was required and explain why not.* |

Note that full information on the approval of the study protocol must also be provided in the manuscript.

# Clinical data

Policy information about clinical studies
All manuscripts should comply with the ICMJE guidelines for publication of clinical research and a completed CONSORT checklist must be included with all submissions.

| Clinical trial registration | *Provide the trial registration number from ClinicalTrials.gov or an equivalent agency.* |
| Study protocol | *Note where the full trial protocol can be accessed OR if not available, explain why.* |
| Data collection | *Describe the settings and locales of data collection, noting the time periods of recruitment and data collection.* |
| Outcomes | *Describe how you pre-defined primary and secondary outcome measures and how you assessed these measures.* |

# Dual use research of concern

Policy information about dual use research of concern

## Hazards

Could the accidental, deliberate or reckless misuse of agents or technologies generated in the work, or the application of information presented in the manuscript, pose a threat to:

| No | Yes | |
|----|-----|---|
| ☐ | ☐ | Public health |
| ☐ | ☐ | National security |
| ☐ | ☐ | Crops and/or livestock |
| ☐ | ☐ | Ecosystems |
| ☐ | ☐ | Any other significant area |

## Experiments of concern

Does the work involve any of these experiments of concern:

No  Yes

☐ | ☐  Demonstrate how to render a vaccine ineffective

☐ | ☐  Confer resistance to therapeutically useful antibiotics or antiviral agents

☐ | ☐  Enhance the virulence of a pathogen or render a nonpathogen virulent

☐ | ☐  Increase transmissibility of a pathogen

☐ | ☐  Alter the host range of a pathogen

☐ | ☐  Enable evasion of diagnostic/detection modalities

☐ | ☐  Enable the weaponization of a biological agent or toxin

☐ | ☐  Any other potentially harmful combination of experiments and agents

# Plants

Seed stocks
*Report on the source of all seed stocks or other plant material used. If applicable, state the seed stock centre and catalogue number. If plant specimens were collected from the field, describe the collection location, date and sampling procedures.*

Novel plant genotypes
*Describe the methods by which all novel plant genotypes were produced. This includes those generated by transgenic approaches, gene editing, chemical/radiation-based mutagenesis and hybridization. For transgenic lines, describe the transformation method, the number of independent lines analyzed and the generation upon which experiments were performed. For gene-edited lines, describe the editor used, the endogenous sequence targeted for editing, the targeting guide RNA sequence (if applicable) and how the editor was applied.*

Authentication
*Describe any authentication procedures for each seed stock used or novel genotype generated. Describe any experiments used to assess the effect of a mutation and, where applicable, how potential secondary effects (e.g. second site T-DNA insertions, mosiacism, off-target gene editing) were examined.*

# ChIP-seq

## Data deposition

☐ Confirm that both raw and final processed data have been deposited in a public database such as GEO.

☐ Confirm that you have deposited or provided access to graph files (e.g. BED files) for the called peaks.

Data access links
*May remain private before publication.*
*For "Initial submission" or "Revised version" documents, provide reviewer access links.  For your "Final submission" document, provide a link to the deposited data.*

Files in database submission
*Provide a list of all files available in the database submission.*

Genome browser session
(e.g. UCSC)
*Provide a link to an anonymized genome browser session for "Initial submission" and "Revised version" documents only, to enable peer review.  Write "no longer applicable" for "Final submission" documents.*

## Methodology

Replicates
*Describe the experimental replicates, specifying number, type and replicate agreement.*

Sequencing depth
*Describe the sequencing depth for each experiment, providing the total number of reads, uniquely mapped reads, length of reads and whether they were paired- or single-end.*

Antibodies
*Describe the antibodies used for the ChIP-seq experiments; as applicable, provide supplier name, catalog number, clone name, and lot number.*

Peak calling parameters
*Specify the command line program and parameters used for read mapping and peak calling, including the ChIP, control and index files used.*

Data quality
*Describe the methods used to ensure data quality in full detail, including how many peaks are at FDR 5% and above 5-fold enrichment.*

Software
*Describe the software used to collect and analyze the ChIP-seq data. For custom code that has been deposited into a community repository, provide accession details.*

# Flow Cytometry

## Plots

Confirm that:

☐ The axis labels state the marker and fluorochrome used (e.g. CD4-FITC).

☐ The axis scales are clearly visible. Include numbers along axes only for bottom left plot of group (a 'group' is an analysis of identical markers).

☐ All plots are contour plots with outliers or pseudocolor plots.

☐ A numerical value for number of cells or percentage (with statistics) is provided.

## Methodology

| | |
|---|---|
| Sample preparation | *Describe the sample preparation, detailing the biological source of the cells and any tissue processing steps used.* |
| Instrument | *Identify the instrument used for data collection, specifying make and model number.* |
| Software | *Describe the software used to collect and analyze the flow cytometry data. For custom code that has been deposited into a community repository, provide accession details.* |
| Cell population abundance | *Describe the abundance of the relevant cell populations within post-sort fractions, providing details on the purity of the samples and how it was determined.* |
| Gating strategy | *Describe the gating strategy used for all relevant experiments, specifying the preliminary FSC/SSC gates of the starting cell population, indicating where boundaries between "positive" and "negative" staining cell populations are defined.* |

☐ Tick this box to confirm that a figure exemplifying the gating strategy is provided in the Supplementary Information.

# Magnetic resonance imaging

## Experimental design

| | |
|---|---|
| Design type | *Indicate task or resting state; event-related or block design.* |
| Design specifications | *Specify the number of blocks, trials or experimental units per session and/or subject, and specify the length of each trial or block (if trials are blocked) and interval between trials.* |
| Behavioral performance measures | *State number and/or type of variables recorded (e.g. correct button press, response time) and what statistics were used to establish that the subjects were performing the task as expected (e.g. mean, range, and/or standard deviation across subjects).* |

## Acquisition

| | |
|---|---|
| Imaging type(s) | *Specify: functional, structural, diffusion, perfusion.* |
| Field strength | *Specify in Tesla* |
| Sequence & imaging parameters | *Specify the pulse sequence type (gradient echo, spin echo, etc.), imaging type (EPI, spiral, etc.), field of view, matrix size, slice thickness, orientation and TE/TR/flip angle.* |
| Area of acquisition | *State whether a whole brain scan was used OR define the area of acquisition, describing how the region was determined.* |

Diffusion MRI        ☐ Used          ☐ Not used

## Preprocessing

| | |
|---|---|
| Preprocessing software | *Provide detail on software version and revision number and on specific parameters (model/functions, brain extraction, segmentation, smoothing kernel size, etc.).* |
| Normalization | *If data were normalized/standardized, describe the approach(es): specify linear or non-linear and define image types used for transformation OR indicate that data were not normalized and explain rationale for lack of normalization.* |
| Normalization template | *Describe the template used for normalization/transformation, specifying subject space or group standardized space (e.g. original Talairach, MNI305, ICBM152) OR indicate that the data were not normalized.* |
| Noise and artifact removal | *Describe your procedure(s) for artifact and structured noise removal, specifying motion parameters, tissue signals and physiological signals (heart rate, respiration).* |

| Volume censoring | *Define your software and/or method and criteria for volume censoring, and state the extent of such censoring.* |
|---|---|

## Statistical modeling & inference

| Model type and settings | *Specify type (mass univariate, multivariate, RSA, predictive, etc.) and describe essential details of the model at the first and second levels (e.g. fixed, random or mixed effects; drift or auto-correlation).* |
|---|---|

| Effect(s) tested | *Define precise effect in terms of the task or stimulus conditions instead of psychological concepts and indicate whether ANOVA or factorial designs were used.* |
|---|---|

Specify type of analysis: ☐ Whole brain ☐ ROI-based ☐ Both

| Statistic type for inference | *Specify voxel-wise or cluster-wise and report all relevant parameters for cluster-wise methods.* |
|---|---|

(See Eklund et al. 2016)

| Correction | *Describe the type of correction and how it is obtained for multiple comparisons (e.g. FWE, FDR, permutation or Monte Carlo).* |
|---|---|

## Models & analysis

| n/a | Involved in the study |
|---|---|
| ☐ | ☐ Functional and/or effective connectivity |
| ☐ | ☐ Graph analysis |
| ☐ | ☐ Multivariate modeling or predictive analysis |

| Functional and/or effective connectivity | *Report the measures of dependence used and the model details (e.g. Pearson correlation, partial correlation, mutual information).* |
|---|---|

| Graph analysis | *Report the dependent variable and connectivity measure, specifying weighted graph or binarized graph, subject- or group-level, and the global and/or node summaries used (e.g. clustering coefficient, efficiency, etc.).* |
|---|---|

| Multivariate modeling and predictive analysis | *Specify independent variables, features extraction and dimension reduction, model, training and evaluation metrics.* |
|---|---|

