## [Peer Review File · Nature]

Manuscript Title: Airborne DNA reveals predictable spatial and seasonal dynamics of fungi

Reviewer Comments & Author Rebuttals

Reviewer Reports on the Initial Version:

Referee #1 (Remarks to the Author):

This study describes global patterns of fungal biodiversity in aerial samples. It describes seasonal patterns of fungal species richness and abundance in different bioclimatic regions and identifies the environmental factors behind these patterns. Overall, I consider this study to be very innovative! It provides new insights for understanding global fungal biodiversity. While most studies on a global scale have so far focused on soil-dwelling fungi, other environments have been largely neglected. Given that virtually all fungal species are at least partially dependent on airborne dispersal, the focus on airborne fungal communities represents a significant step forward in understanding fungal biodiversity and their communities. In addition, data from airborne fungal communities can help us better understand the dispersal capabilities of fungal species.

Major comments

It should be noted that the composition of airborne fungal communities appears to be slightly skewed towards wood-decaying saprotrophic and plant pathogenic fungi, while ectomycorrhizal fungi, for example, are probably under-represented compared to soil fungal communities. Results based on aerial fungal communities should therefore be interpreted as another piece of the puzzle in the global picture of fungal biodiversity rather than as an overarching methodology for understanding fungal biogeography and macroecology. This aspect should be included in the discussion (lines 430-453).

Considering the methodology, the authors used routine metabarcoding methods for identification of fungal community composition. To account for PCR and sequencing errors, they decided to denoise the ITS sequencing data using the approach which includes construction of the amplicon sequence variants (ASVs). Personally, I do not think that this approach compromise the results, but I would like to stress that the ASVs approach is rather suboptimal, and not recommended by fungal biologists (e.g., Kauserud 2023 Fungal Ecology DOI: 10.1016/j.funeco.2023.101274) for ITS based identification of fungal taxa. Besides this, I find the methods for identification of fungal taxonomy and ecology appropriate. I would particularly like to stress very well performed identification of putative fungal taxa on higher taxonomical levels than species or genera, which was done by Protax, which enables identification of putative fungal taxa even from largely under sampled regions, such as tropics.

Although I consider the introduction to be quite thorough and well-structured, I would still recommend giving a little more information on at least the important ecological groups of fungi. For example, the authors mention the specific distribution of ectomycorrhizal fungal richness, which does not follow the most common negative latitudinal pattern, but do not particularly explain what ectomycorrhizal fungi are. If this manuscript was being considered for publication in a discipline-specific scientific journal (e.g., *New Phytologist*, *The ISME Journal*), this would not matter so much, but I think more information is beneficial to a more general audience of Nature. In addition, the fungal guild is also mentioned as one of the two key traits to determine whether species respond to

climatic and seasonal factors based on their traits.

I do not understand, why arbuscular mycorrhizal, ericoid mycorrhizal and endophytes (not clear if root and shoot endophytes together) were considered as one functional guild. These groups have very little in common and very likely other environmental factors will be affecting distribution of species with these ecologies. Considering that the study focuses primarily on air-borne fungi, I would recommend treating at least the shoot endophytes independently, as air represents their prevailing distribution vector.

Minor comments:

Authors decided to present their dataset in a separated paper, which is currently under revision in Scientific Data. I indeed think that their dataset is so complex and valuable that it is fully justified to give more space to the dataset itself, I think that at least short summary of molecular and bioinformatic methods would be beneficial for this “research” manuscript. I guess that readers would appreciate that.

The authors expected that spatial differentiation of airborne spores is less pronounced than previously reported in soil-based studies, as fungal spores mix more readily in air compared to soil. Although I agree with this statement, it should be noted that the spore samples were collected one meter above ground and often in a forest environment with probably rather limited air flow. The global aspect of this study makes this study very interesting. The only disadvantage is relatively imperfect coverage of temperate region, where only one two out of 17 sites are outside of Europe and only a single sample, assigned to temperate bioclimatic zone, originates from the Southern Hemisphere.

I was wondering how well were the air-borne fungal communities described. In the case of soil samples, we are dealing with very complex and species-rich fungal communities where we hardly achieve saturation of the species accumulation curve. However, I would expect that even relatively shallow sequencing (say 5,000 sequences per sample) can potentially detect almost all species in a 24 m³ air sample, right? Can you provide more information about the completeness of the identified fungal communities?

The authors focused their analyses on the effects of climatic conditions and geographic distance on fungal communities in air samples. In case of soil fungal communities, large part of their variation on large spatial scales is explained by biome. I was a bit surprised that the effect of vegetation type on composition of air-borne fungal communities was not covered in the analysis. Was there any particular reason why?

Just a note: According to the Figure 1A Taxonomic composition, I would assume that almost no arbuscular mycorrhizal fungi were detected, as no Mucoromycota or Glomeromycota are displayed in the graph, right?

The statistical analyses are very well described, and I found no flaws in them.

Referee #2 (Remarks to the Author):

The authors collected an impressive dataset to investigate spatiotemporal dynamics in airborne fungal spores across 47 sampling locations. They used this dataset to look at spatial and seasonal patterns in spore diversity and community composition. While there have been a reasonably large number of studies investigating fungal diversity in air samples collected from individual locations, this study is unique in its scope. I do have some concerns about this study that are detailed below.

- I was surprised the authors put the details of their methods in another paper (Ovaskainen et al.) instead of just including the methodological details here. This was a bit frustrating as I essentially had to review two manuscripts as the methodological information is important and I expect that readers will feel similarly. I know that space is limited, but some details on how the sampling was conducted and how the sequence data were generated/processed would be very valuable to include in this manuscript.

- I have one very important concern about their methods. After reading the Ovaskainen et al. companion paper that describes the molecular methods – it does not appear that any field or laboratory controls were included to check for potential contamination. This is very important as contamination, either during field collection or sample processing, is hard to avoid when dealing with low biomass samples such as the ones collected for this study (even for fungi). It does seem like the spore traps were regularly cleaned (as mentioned in the companion paper) but the cleaning process seemed a bit inconsistent between sites. From other DNA-based microbial studies in similarly low biomass systems, we know that the specific contaminants introduced can often be highly site specific, i.e. different sites could have different amounts/types of potential fungal DNA contaminants introduced during sample collection and initial processing (contamination that could strongly impact their spatial analyses). I do not see any mention of any negative controls being analyzed alongside the samples at any of the steps (field collection, DNA extraction, PCR) and this is very problematic. In my opinion the manuscript is not publishable unless the authors can show that they included the appropriate controls and can confirm that their results were unaffected by any contamination introduced during the field collection process or in the laboratory post-collection.

- Why do the authors think MAT was more important than MAP in structuring the airborne fungal assemblages? One possibility is that MAP is such a coarse metric that it does not effectively indicate moisture availability (soil moisture deficit is not necessarily equivalent to MAP). It would be useful to calculate an aridity index, instead of MAP, as I would expect that the differences in aridity levels across sites are a better predictor of the fungal assemblage composition than MAP alone.

- Lines 350-352: Is the over-representation of ectomycorrhizal species and lichens in the temperate regions a product of taxa from temperate regions simply being better characterized?

- I found the presentation of the results regarding seasonality in fungal guilds and the associated phylogenetic signals to be quite confusing. Part of this confusion stems from the fact that Pagel's lambda values are difficult to interpret as the values do not necessarily indicate at what level of phylogenetic/taxonomic resolution the signals are apparent. I would recommend the authors include a few more examples to illustrate these results and make them less abstract.

- Lines 434-437: Where are the data to support these conclusions about the proportional representation of fungi from different trophic guilds? I assume the authors are referring to Figure 2, panel B here?

- For Figure 3, panels A, B, D, and E – the raw data need to be shown – or at least examples of what the raw data look like for selected sites. I'm very skeptical that the patterns are anywhere close to as clean as those shown here based on the lines from the linear mixed models. Readers should be aware of how well these models do, or do not, reflect the actual data.

- I would recommend the authors go into more detail on why they think MAT is so important in explaining the site-specific patterns observed. Is it because MAP is not a good indicator of climatic conditions relevant to fungi? If so, are there other indices of climatic conditions that would be more appropriate to use (see comment above).

- Were the data rarefied before estimating fungal richness? This is important as I assume the number of sequence reads obtained per sample is highly variable. I don't see any mention of this in either manuscript. More generally, it would be useful to include a summary of the number of reads obtained across the samples (non-spike) and what percent of samples were discarded due to insufficient sequencing depth.

Referee #3 (Remarks to the Author):

Dear editor, dear authors, I will perform a critical analysis point by point of each part of the manuscript with title "Airborne DNA metabarcoding reveals that fungi follow predictable spatial and seasonal dynamics at the global scale". This manuscript presents a significant contribution to the field of mycology and global biodiversity studies by leveraging airborne DNA metabarcoding to explore the spatial and seasonal dynamics of fungal communities across various climatic zones. The study stands out for its global scale and the application of advanced molecular and bioinformatic techniques to address a gap in current understanding of fungal distributions. The findings that fungal diversity exhibits predictable patterns related to climate, latitude, and seasonality are compelling and advance our knowledge of ecological and evolutionary processes shaping microbial life on a global scale. The rigorous methodology, comprehensive analysis, and clear presentation of results support the study's conclusions and highlight its potential to inform future research in ecology, conservation, and climate change studies. Overall, the manuscript is well-positioned to make a valuable impact on its field, subject to any specific revisions that might further clarify and enhance its contributions. The title is both informative and precise, clearly conveying the scope and findings of the study.

I do not have major concerns about the manuscript, just some suggestions. While the global scale of the study is impressive, there appears to be uneven geographic distribution of sampling sites, with potential gaps in some biomes. I understand the limitations of space required by the journal but a brief but precise acknowledgment of the study's limitations and suggestions for future research directions can significantly enhance the contribution. This might include a sentence or two on the geographic coverage of samples, potential biases in DNA metabarcoding, or areas where further methodological development could yield deeper insights.

About the methodology, I appreciate the manuscript's detailed explanation of the optimized method for monitoring fungal spores. However, given the known presence of airborne hyphae in environmental samples, a verification step using an optical microscope for some samples, alongside a comparison with a Hirst sampler, could enhance the robustness of the findings. If such verification is not feasible, I recommend moderating the claims regarding the specificity of the sampling method to fungal spores, to acknowledge the potential for capturing other fungal structures.

As minor concerns, ensure that taxonomic names, locations, and technical terms are consistently capitalized and italicized as per the relevant scientific convention. This includes genus and species names in italics and consistent use, for example see Table 1 and Figure 2A.

I am not native speaker, but I found no one typographical error or mistakes referring grammar and punctuation. Neither related with references. Figures and tables have good resolution and are high quality.

In conclusion, this manuscript presents a well-conducted and valuable study that significantly contributes to our understanding of the ecology of airborne fungal signals. The authors have addressed a relevant question with rigorous methodology and clear presentation of their findings.

Referee #3 (Remarks on code availability):

I am unable to download the R scripts, even with the provided token. It seems that some additional configuration is needed. Data are public but restricted to users with access.

Author Rebuttals to Initial Comments:

Referee: #1

R1.1. This study describes global patterns of fungal biodiversity in aerial samples. It describes seasonal patterns of fungal species richness and abundance in different bioclimatic regions and identifies the environmental factors behind these patterns. Overall, I consider this study to be very innovative! It provides new insights for understanding global fungal biodiversity. While most studies on a global scale have so far focused on soil-dwelling fungi, other environments have been largely neglected. Given that virtually all fungal species are at least partially dependent on airborne dispersal, the focus on airborne fungal communities represents a significant step forward in understanding fungal biodiversity and their communities. In addition, data from airborne fungal communities can help us better understand the dispersal capabilities of fungal species.

RESPONSE: We thank the reviewer for their encouraging comments on our manuscript.

Major comments

R1.2. It should be noted that the composition of airborne fungal communities appears to be slightly skewed towards wood-decaying saprotrophic and plant pathogenic fungi, while ectomycorrhizal fungi, for example, are probably under-represented compared to soil fungal communities. Results based on aerial fungal communities should therefore be interpreted as another piece of the puzzle in the global picture of fungal biodiversity rather than as an overarching methodology for understanding fungal biogeography and macroecology. This aspect should be included in the discussion (lines 430-453).

RESPONSE: Excellent point, we now have added this perspective to the Discussion.

R1.3. Considering the methodology, the authors used routine metabarcoding methods for identification of fungal community composition. To account for PCR and sequencing errors, they decided to denoise the ITS sequencing data using the approach which includes construction of the amplicon sequence variants (ASVs). Personally, I do not think that this approach compromise the results, but I would like to stress that the ASVs approach is rather suboptimal, and not recommended by fungal biologists (e.g., Kausrud 2023 Fungal Ecology DOI: 10.1016/j.funeco.2023.101274) for ITS based identification of fungal taxa. Besides this, I find the methods for identification of fungal taxonomy and ecology appropriate. I would particularly like to stress very well performed identification of putative fungal taxa on higher taxonomical levels than species or genera, which was done by Protax, which enables identification of putative fungal taxa even from largely under sampled regions, such as tropics.

RESPONSE: We thank the reviewer for their kind comments on the merits of using Protax for species identification, and we consider that this approach overcomes some of the limitations of the ASV approach. Kausrud (2023) notes that there is an “overestimation” problem with applying the ASV approach to ITS data; and this is because the diversity can be considerably overestimated if each ASV is treated as a separate species without accounting for high levels of intraspecific variation. Having said that, we note that our pipeline does *not* consider each ASV as a separate species, but simply

uses the ASV approach as the first denoising step. As explained in more detail in the data paper by Ovaskainen et al. (manuscript uploaded with this contribution), we apply probabilistic taxonomic placement using Protax-fungi *after* the ASV step. With this approach, we assign ASVs to taxa at ranks from phylum to species. Finally, we use a new constrained clustering approach to group ASVs into species-level operational taxonomic units. This step is explicitly guided by the taxonomic annotations from Protax-fungi. During these later steps, many ASVs are merged into one putative species, thus reducing the risk of overestimation of diversity.

In response to the reviewer's concern, we now explain a summary of our bioinformatics pipeline in the present manuscript, and in full detail in the data paper. In both cases, we now also refer to Kausserud (2023) to note why the ASV approach should be applied cautiously to ITS data.

In the brief methods section of the present manuscript we mention as follows:

"Due to the unsuitability of using ITS-based ASVs as proxies for species (Kausserud 2023), we developed a taxonomically-guided clustering approach to form species-level OTUs. We performed a probabilistic taxonomic placement..."

We have added the following new motivating summary of the bioinformatic steps to the data paper:

"Due to frequent intraspecific sequence variants for the ITS region, ITS-based ASVs are not suitable proxies for species (Kausserud 2023). Consequently, we developed a taxonomically-guided clustering approach using the taxonomic annotations from Protax-fungi to group ASVs into approximately species-level OTUs. Our approach also groups sequences, including those without existing taxonomic annotations, into clusters approximating each taxonomic rank. First, we calculated optimum single-linkage clustering thresholds for..."

R1.4. Although I consider the introduction to be quite thorough and well-structured, I would still recommend giving a little more information on at least the important ecological groups of fungi. For example, the authors mention the specific distribution of ectomycorrhizal fungal richness, which does not follow the most common negative latitudinal pattern, but do not particularly explain what ectomycorrhizal fungi are. If this manuscript was being considered for publication in a discipline-specific scientific journal (e.g., New Phytologist, The ISME Journal), this would not matter so much, but I think more information is beneficial to a more general audience of Nature. In addition, the fungal guild is also mentioned as one of the two key traits to determine whether species respond to climatic and seasonal factors based on their traits.

RESPONSE: We thank the reviewer for this constructive suggestion. We have now added key information about the roles of different fungal groups to the Introduction:

"Fungi are among the most diverse and ecologically important living organisms. They mediate crucial processes in terrestrial ecosystems as decomposers of dead tissues (saprotrophs), mutualistic partners (ectomycorrhizal, ericoid, endophytic, and lichenized fungi), and pathogens, of almost all terrestrial multicellular organisms..."

While this is admittedly very short, we did not find it possible to add more information given the strict length restrictions. We believe the other changes have made the Introduction more accessible to readers who are not familiar with fungal ecology.

R1.5. I do not understand, why arbuscular mycorrhizal, ericoid mycorrhizal and endophytes (not clear if root and shoot endophytes together) were considered as one functional guild. These groups have very little in common and very likely other environmental factors will be affecting distribution of species with these ecologies. Considering that the study focuses primarily on air-borne fungi, I would recommend treating at least the shoot endophytes independently, as air represents their prevailing distribution vector.

RESPONSE: These groups were analyzed together mainly due to the small sample size, especially for arbuscular mycorrhizal taxa. Nonetheless, we agree with the argument of the reviewer and have now analyzed them separately as suggested. This analysis showed that ericoid mycorrhizal species peak at high latitudes, in contrast to endophytes (as could indeed be expected). Thus, the separation offered by the reviewer revealed an interesting ecological pattern. Unfortunately, the arbuscular mycorrhizal species were too rare (n=14) in our data to include them as a separate group in these analyses so they were excluded from these group-specific analyses. We have also added to the Methods that the limited coverage of arbuscular mycorrhiza can be due to methodological limitations: "However, we note that for some fungal taxa other markers are better suited, such as the nuclear SSU rRNA gene fragment for arbuscular mycorrhiza (Öpik et al. 2013)."

Minor comments

R1.6. Authors decided to present their dataset in a separated paper, which is currently under revision in Scientific Data. I indeed think that their dataset is so complex and valuable that it is fully justified to give more space to the dataset itself, I think that at least short summary of molecular and bioinformatic methods would be beneficial for this "research" manuscript. I guess that readers would appreciate that.

RESPONSE: We fully agree and have followed the suggestion by adding the following summary of the molecular and bioinformatics methods:

"Study design, DNA extraction, sequencing, and bioinformatics

For full details on study design and sample collection, DNA extraction and sequencing, bioinformatic processing, as well as technical data validation, see Ovaskainen et al (submitted). Here we summarize these steps.

The study design consists of 47 sampling sites, each equipped with a cyclone sampler (Burkard Cyclone Sampler for Field Operation, Burkard Manufacturing Co Ltd; <http://burkard.co.uk/product/cyclone-sampler-for-field-operation>). The sampling sites were selected to represent local natural environments, where intensive, continuous sampling was possible. The cyclone samplers collected particles >1µm in size from the air directly into a sterile Eppendorf vial, with average air throughput of 23.8 m³ during each 24-hour sampling period. Prior to the start of our global sampling, a field test was performed to evaluate the quantity of fungal DNA collected over different time frames. We also included field blanks handled with and without gloves, in which the sampler was not activated, and the Eppendorf vials were removed after one minute and sealed. As a result of the field tests, we selected a 24-hour sampling period and instructed the participants to handle the samples with gloves and to clean the cyclone parts monthly.

We amplified the ITS2 region using the polymerase chain reaction (PCR) for 20 cycles with fusion primers ITS_S2F(Chen et al., 2010), ITS3, and ITS4(White et al., 1990) tailed with Illumina adapters, and sequenced them on Illumina MiSeq. In the MiSeq runs, we included two sets of negative control samples, introduced at the DNA extraction step and at the PCR step, respectively. Of the 99 total negative control samples, 89% (88 samples) did not yield any reads of fungal origin. The remaining 9 negative control samples included a few fungal reads (relative to the study samples) of relatively common OTUs, suggesting infrequent cross-contamination. To test the robustness of the results with respect to such cross-contamination, we repeated two of the main analyses (variation in species richness and joint species distribution modelling) with data that we purposely contaminated with the observed level of cross-contamination. To do so, we added to the OTU reads of each field sample the OTU reads of a randomly selected negative control sample. We replicated the cross-contamination simulation for ten independent replicates, with results being almost identical to results obtained to the original data (see *Supporting Information*).

To quantify the amount of fungal DNA, we applied a spike-in approach, and we converted the ratio of the non-spike vs. spike-sequences into semi-quantitative estimates of DNA amount (Ovaskainen et al., 2020). Demultiplexed paired-end reads were trimmed, denoised, and chimera checked using Cutadapt version 4.2 (Martin, 2011), DADA2 version 1.18.0(Callahan et al., 2016), and VSEARCH version 2.22.1(Rognes et al., 2016). As reference database, we used Sanger sequences from the UNITE v9 database (Abarenkov et al., 2024) supplemented with the synthetic spike sequences. Sequences representing non-spike amplicon sequence variants (ASV (Callahan et al., 2017)) were aligned between the ITS3 and ITS4 primer sites. Discarding sequences that did not match the full length of the model, or which had a bit score less than 50 resulted in a 65,912 ASVs x 2,768 samples matrix of read abundance.

Due to the unsuitability of using ITS-based ASVs as proxies for species (Kausarud, 2023), we developed a taxonomically-guided clustering approach to form species-level OTUs. We performed a probabilistic taxonomic placement of the ASVs with Protax-fungi (Abarenkov et al., 2018) with a 90% probability threshold. Additionally, sequences whose best match to UNITE Sanger sequences was to a kingdom other than Fungi were annotated as potential non-fungi. We applied constrained clustering by first forming cluster cores by the ASVs which had been assigned to taxa by Protax-fungi. We then matched the unassigned ASVs to the closest cluster core using optimized sequence similarity thresholds. Finally, remaining unclustered ASVs were clustered using de novo single-linkage clustering. These de novo clusters were assigned to placeholder taxonomic names of the form “pseudo{rank}_{number}”. The final result of this process was a 27,954 species-level OTUs x 2,768 samples read abundance matrix, along with taxonomic annotations at each rank from phylum to species, including pseudotaxon placeholders.

The mean sequencing depth (total number of fungal and spike sequences) among the samples was 86,845 sequences per sample. Based on rarefaction analyses presented in Ovaskainen et al. (submitted), we discarded samples that did not contain at least 10,000 sequencing reads, representing 1.8% of the samples. To avoid losing some OTUs detected in the most diverse samples, we controlled for variation in sequencing depth by statistical means rather than using rarefied values (McMurdie & Holmes, 2014).”

R1.7. The authors expected that spatial differentiation of airborne spores is less pronounced than previously reported in soil-based studies, as fungal spores mix more readily in air compared to soil. Although I agree with this statement, it should be noted that the spore samples were collected one meter above ground and often in a forest environment with probably rather limited air flow.

RESPONSE: We have added this important remark to the Introduction as “although the samples were collected close to the ground, and often within habitats with limited air flow compared to open areas.”

R1.8. The global aspect of this study makes this study very interesting. The only disadvantage is relatively imperfect coverage of temperate region, where only one two out of 17 sites are outside of Europe and only a single sample, assigned to temperate bioclimatic zone, originates from the Southern Hemisphere.

RESPONSE: We appreciate the reviewer’s praise of the global aspect of the study. We agree that having a more balanced global representation of the different bioclimatic zones would have been better. Nonetheless, this was not feasible – given that the study was based on collaboration with research groups who volunteered to conduct intensive sampling at their study sites. We note that we made a substantial effort to locate such collaborators from outside Europe, especially from the Global South which is unfortunately under-represented in global biodiversity studies. Even though the resulting sampling was clearly most intensive in Europe, we are still satisfied that we achieved a truly global scope, including representation of otherwise little studied areas. However, we fully agree with the reviewer and when describing the data, we now explicitly mention the over-representation of the temperate region as a limitation.

R1.9. I was wondering how well were the air-borne fungal communities described. In the case of soil samples, we are dealing with very complex and species-rich fungal communities where we hardly achieve saturation of the species accumulation curve. However, I would expect that even relatively shallow sequencing (say 5,000 sequences per sample) can potentially detect almost all species in a 24 m³ air sample, right? Can you provide more information about the completeness of the identified fungal communities?

RESPONSE: To address this question, we have added the following new section to the data paper (Ovaskainen et al.; enclosed):

“Sufficiency of sequencing depth

The mean sequencing depth among the samples was 86,845, and the median sequencing depth was 79,396. We recommend conducting analyses with samples that obtained at least 10,000 sequencing reads, which corresponds to discarding 50 samples and thus 1.8% of the samples (Fig. 3A). If rarefying all samples to 10,000 sequence reads, a minor loss of species-level OTU richness is observed for the most diverse samples (Fig. 3B). Nonetheless, even the most diverse samples were likely sequenced to an adequate depth, as illustrated by the well-saturating rarefaction curves (Fig. 3C).

Figure 3. Results illustrating the sufficiency of sequencing depth, i.e., the total number of sequencing reads (including fungal and spike reads) obtained for each sample. Panel A shows the distribution of sequencing depth among the samples, with the dashed vertical line corresponding to 10,000 sequence reads that we recommend using as a threshold for including a sample for analyses. Panel B shows the decrease in the number of species-level OTUs if rarefying all samples to 10,000 sequence reads. Panel C shows rarefaction curves for all samples that included at least 10,000 sequence reads.”

In the present manuscript, we summarized this in the new Methods section as follows:

“The mean sequencing depth (total number of fungal and spike sequences) among the samples was 86,845 sequences per sample. Based on rarefaction analyses presented in Ovaskainen et al. (submitted), we discarded samples that did not contain at least 10,000 sequencing reads, representing 1.8% of the samples. To avoid losing some of the OTUs detected in the most diverse samples, we controlled for variation in sequencing depth by statistical means rather than using rarefied values.”

R1.10. The authors focused their analyses on the effects of climatic conditions and geographic distance on fungal communities in air samples. In case of soil fungal communities, large part of their variation on large spatial scales is explained by biome. I was a bit surprised that the effect of vegetation type on composition of air-borne fungal communities was not covered in the analysis. Was there any particular reason why?

RESPONSE: While we agree that the effect of biome (and many other predictors) would have been interesting to test, we have now added an explanation of why we did not find it possible from the statistical point of view:

“The reason for including only a small number of site-specific variables in the analysis is that, while the study is global in scope, it includes only 47 sites. The data thus hold limited information to statistically disentangle the effects of many spatially-varying covariates. Instead, the main strength of the study lies in its high temporal replication, which allowed us to identify effects of the spatiotemporal covariates, such as seasonality.”

In comparison, the global soil fungal data have a higher level of spatial replication, but they lack the temporal aspect. We note that while n=47 may sound large enough for disentangling the effects of a

reasonably large number of covariates, these data points are not independent due to spatial autocorrelation. In other words, many locations in Europe represent similar conditions in terms of their biome and climatic conditions, and attributing the effects to either factor would then be questionable. As temperature and precipitation largely determine the biomes, we decided to use them instead. In response, we have now discussed the point that studies on soil fungal communities have found the biome to explain a major part of the variation:

“Likewise, previous studies on soil fungal communities have found that biomes, as defined based on MAT and MAP, explain a major part of their global distributions (Tedersoo et al. 2014).”

R1.11. Just a note: According to the Figure 1A Taxonomic composition, I would assume that almost no arbuscular mycorrhizal fungi were detected, as no Mucoromycota or Glomeromycota are displayed in the graph, right?

RESPONSE: The assumption of the reviewer is correct: using the classification method described in the manuscript, which is based on the Aguilar-Trigueros et al. database, only 14 species-level OTUs were classified to arbuscular mycorrhizal fungi.

R1.12. The statistical analyses are very well described, and I found no flaws in them.

RESPONSE: We thank the reviewer for this positive assessment.

Referee #2

R2.1. The authors collected an impressive dataset to investigate spatiotemporal dynamics in airborne fungal spores across 47 sampling locations. They used this dataset to look at spatial and seasonal patterns in spore diversity and community composition. While there have been a reasonably large number of studies investigating fungal diversity in air samples collected from individual locations, this study is unique in its scope. I do have some concerns about this study that are detailed below.

RESPONSE: We thank the reviewer for this overall positive evaluation.

R2.2. I was surprised the authors put the details of their methods in another paper (Ovaskainen et al.) instead of just including the methodological details here. This was a bit frustrating as I essentially had to review two manuscripts as the methodological information is important and I expect that readers will feel similarly. I know that space is limited, but some details on how the sampling was conducted and how the sequence data were generated/processed would be very valuable to include in this manuscript.

RESPONSE: The same issue was identified by reviewer #1. Thus, we thank you for pointing out that presenting more methodological information in this manuscript is necessary. In response, we have now included key information about sampling, DNA-extraction, sequencing, and bioinformatics in the Methods. We note that we originally aimed to publish both the data paper and the present paper as a single study, but that given the length restrictions, this resulted in too many compromises. For this reason, we decided to publish the data paper separately, and to include full details there. We feel that this solution will be beneficial to the re-analysis and further use of the data by the scientific community, as the data paper can include in-depth information on every methodological choice made.

R2.3. I have one very important concern about their methods. After reading the Ovaskainen et al. companion paper that describes the molecular methods – it does not appear that any field or laboratory controls were included to check for potential contamination. This is very important as contamination, either during field collection or sample processing, is hard to avoid when dealing with low biomass samples such as the ones collected for this study (even for fungi). It does seem like the spore traps were regularly cleaned (as mentioned in the companion paper) but the cleaning process seemed a bit inconsistent between sites. From other DNA-based microbial studies in similarly low biomass systems, we know that the specific contaminants introduced can often be highly site specific, i.e. different sites could have different amounts/types of potential fungal DNA contaminants introduced during sample collection and initial processing (contamination that could strongly impact their spatial analyses). I do not see any mention of any negative controls being analyzed alongside the samples at any of the steps (field collection, DNA extraction, PCR) and this is very problematic. In my opinion the manuscript is not publishable unless the authors can show that they included the appropriate controls and can confirm that their results were unaffected by any contamination introduced during the field collection process or in the laboratory post-collection.

RESPONSE: We are most grateful to the reviewer for pointing out this oversight in our previous reporting of the Results. We did indeed include a total of 99 negative controls in the laboratory

phase, and an added a set of field blanks, but simply failed to describe them in the Ovaskainen et al. manuscript. To rectify this issue, we have now included the following new section in the Ovaskainen et al. manuscript:

“Field tests and negative controls

The median DNA amount measured by RT-PCR in the seven 24-hour test samples was 14 fg of DNA. The median DNA content measured in 1-hour samples was 8 fg, and the median for 10-minute samples, as well as for field blanks handled without gloves, were less than 3 fg. The median DNA quantity measured in the field blanks handled with gloves and the extraction blanks were approximately 0.7 fg, and the DNA quantity in the PCR blank was approximately 0.1 fg (Fig. 2A). Although these values were standardized using genomic DNA extracted from yeast, they cannot be directly translated to other fungi due to varying genome size and ITS copy number, it is informative that actual 24-hour samples had almost 5 times more ITS copies than blank samples handled without gloves, and twenty times more than blank samples handled with gloves. In the actual study, all samples were handled with gloves.

Of the 99 negative controls, 89% (88 samples) did not yield any reads of fungal origin at the end of the bioinformatic analysis. For all sequencing runs, at least one negative control sample contained 0 fungal reads, indicating that the reagents were uncontaminated. The 9 negative control samples which did include fungal reads yielded fewer fungal reads than the study samples (Fig. 2B), and, in most cases, these reads belonged to only one or two OTUs. OTUs found in negative control samples were all relatively common in the study. They were no more common in the sequencing runs which contained the negative controls than in other sequencing runs. This suggests that the most likely source of these reads was infrequent cross-contamination from study samples to negative controls. Among the negative controls, sample CCDB-35071NEGPCR2 yielded the highest read count: 2668 fungal reads. All 18 OTUs detected in this sample were also found in sample COR_41A, with abundances 7-60 times as high as in the negative control. Samples CCDB-35071NEGPCR2 and COR_41A were processed in the same sequencing run, indicating that the sample COR_41A was likely the source of cross-contamination.

Figure 2. Results from field tests and negative controls. Panel A shows DNA concentration in the field test samples, based on either 24-hr sampling, 1-hr sampling, or 10-min sampling, PCR blanks, extraction blanks, and field blanks handled with and without gloves. Panel B shows the distributions

of the number of fungal reads per sample, based on either field samples (green bars), field blanks (blue bars) or lab blanks (red bars). Note the logarithmic scale in the x-axis.”

In the present manuscript, we briefly summarize these findings in the new Methods section as follows:

“Prior to the start of our global sampling, a field test was performed to evaluate the quantity of fungal DNA collected over different time frames. We also included field blanks handled with and without gloves, in which the sampler was not activated, and the Eppendorf vials were removed after one minute and sealed. As a result of the field tests, we selected a 24-hour sampling period and instructed the participants to handle the samples with gloves and to clean the cyclone parts monthly.”

“In the MiSeq runs, we included two sets of negative control samples, introduced at the DNA extraction step and at the PCR step, respectively. Of the 99 total negative control samples, 89% (88 samples) did not yield any reads of fungal origin. The remaining 9 negative control samples included a few fungal reads (relative to the study samples) of relatively common OTUs, suggesting infrequent cross-contamination. To test the robustness of the results with respect to such cross-contamination, we repeated two of the main analyses (variation in species richness and joint species distribution modelling) with data that we purposely contaminated with the observed level of cross-contamination, the results being almost identical to results obtained to the original data (see *Supporting Information*).”

In the Supporting Information, we now have included a major new section that shows that the results of our statistical analyses are highly robust with respect to contamination. As that new section is seven pages long (including figures and tables), we do not replicate it here.

R2.4. Why do the authors think MAT was more important than MAP in structuring the airborne fungal assemblages? One possibility is that MAP is such a coarse metric that it does not effectively indicate moisture availability (soil moisture deficit is not necessarily equivalent to MAP). It would be useful to calculate an aridity index, instead of MAP, as I would expect that the differences in aridity levels across sites are a better predictor of the fungal assemblage composition than MAP alone.

RESPONSE: We thank the reviewer for this constructive suggestion. In response, we have included the suggested analyses on the influence of aridity, in addition to the influences of MAT, MAP, and mean wind speed. Thus, all spatial patterns are now related to these four predictors. In our results, aridity turned out to have an explanatory power similar to that of MAP. Hence, MAT remained by far the predictor with the highest explanatory power. We agree that aridity is a very valuable addition to our paper.

R2.5. Lines 350-352: Is the over-representation of ectomycorrhizal species and lichens in the temperate regions a product of taxa from temperate regions simply being better characterized?

RESPONSE: This is indeed a possibility, and we now mention this in the text:

“While the higher diversity of these fungal groups at higher latitudes could be related to greater knowledge gaps of their diversity in the tropics, this result could also reflect the distribution and diversity of their host species (Tedersoo and Nara, 2009).”

Furthermore, as we now stress in the discussion, to minimize the possibility of such an artefact, we borrowed information among taxonomic levels for the functional classifications. In other words, where information was available at the genus and family levels, we assigned the same classification to taxa for which species-level information was missing. Thus, many of the tropical species-level OTUs were classified as ectomycorrhizal species or lichens, even if we could not identify them to a species level. Borrowing information from genus and family levels clearly comes with the trade-off of adding noise where species within the genus or family actually belong to different functional classes. Nonetheless, we made this choice as a compromise between minimizing bias (by only including not only the minority of OTUs reliably classified to species but also genus- or family-level classifications) and minimizing the noise of false classifications (by not borrowing information from the order level). In terms of the reviewer's specific concern that "*the over-representation of ectomycorrhizal species and lichens in the temperate regions [may be] a product of taxa from temperate regions simply being better characterized*", we note that the same concern will apply to other taxa, too. Not only ectomycorrhizal species and lichens are better characterized in temperate than tropical regions, but the same holds true for essentially all species groups. As we did not find such a pattern for the other species groups, it does not appear likely that the pattern found for ectomycorrhizal, lichenized and ericoid mycorrhizal fungi would be solely an artefact.

R2.6. I found the presentation of the results regarding seasonality in fungal guilds and the associated phylogenetic signals to be quite confusing. Part of this confusion stems from the fact that Pagel's lambda values are difficult to interpret as the values do not necessarily indicate at what level of phylogenetic/taxonomic resolution the signals are apparent. I would recommend the authors include a few more examples to illustrate these results and make them less abstract.

RESPONSE: We thank the reviewer for challenging us to clarify our presentation. In response, we have now extended the original figure with the distributions of climatic and seasonal sensitivities (i.e., those variables that showed a phylogenetic pattern) for an additional illustration focusing on orders represented by at least 10 species. Accordingly, we have expanded the text to highlight that *Agaricales* and *Helotiales* showed low climatic sensitivity, *Polyporales* and *Erysiphales* high seasonal sensitivity, and *Agaricales*, *Tremellales* and *Chaetothyriales* low seasonal sensitivity.

R2.7. Lines 434-437: Where are the data to support these conclusions about the proportional representation of fungi from different trophic guilds? I assume the authors are referring to Figure 2, panel B here?

RESPONSE: The reviewer is right that we refer here to Fig. 2B – a clarification which we have now added to the text.

R2.8. For Figure 3, panels A, B, D, and E – the raw data need to be shown – or at least examples of what the raw data look like for selected sites. I'm very skeptical that the patterns are anywhere close to as clean as those shown here based on the lines from the linear mixed models. Readers should be aware of how well these models do, or do not, reflect the actual data.

RESPONSE: The reviewer is absolutely right that the raw data are not as clean as the means predicted by the models. We have now illustrated the amount of variation in the raw data by plotting them in the backgrounds of panels ABDE, as well as in the corresponding figures in the

Supporting Information.

R2.9. I would recommend the authors go into more detail on why they think MAT is so important in explaining the site-specific patterns observed. Is it because MAP is not a good indicator of climatic conditions relevant to fungi? If so, are there other indices of climatic conditions that would be more appropriate to use (see comment above).

RESPONSE: As noted above (response to R2.4), we have now added aridity to the analyses, as suggested by the reviewer #2. We note that MAT remains the predictor with the highest explanatory power. One possibility for why MAT proves so important for fungi is because it correlates with biome and hence with plant distributions, and because many fungi are plant host-dependent. We have added the following to the Discussion:

“Likewise, previous studies on soil fungal communities have found that biomes, as defined based on MAT and MAP, explain a major part of their global distributions (Tedersoo et al. 2014).”

R2.10. Were the data rarefied before estimating fungal richness? This is important as I assume the number of sequence reads obtained per sample is highly variable. I don't see any mention of this in either manuscript. More generally, it would be useful to include a summary of the number of reads obtained across the samples (non-spike) and what percent of samples were discarded due to insufficient sequencing depth.

RESPONSE: The data were not rarefied. Instead, we corrected for variation in sequencing depth by including the log-transformed number of sequence reads as a covariate. This was explained in Methods section: “Model CS1: Null model. The null model does not include any ecological predictors as fixed effects but includes log(sequencing depth) for the species richness model. To account for the study design with multiple samples from the same locations, the null model includes the site as a random intercept.” The other models extend the null model and hence also include log(sequencing depth) as a covariate. We have now followed the reviewer’s suggestion and added information about the number of reads obtained for each sample. The full details are shown in the revised version of the data paper (provided as supplementary file), where the new Fig. 3 shows the distribution of the number of sequencing reads, the influence of rarefying the data to 10,000 sequence reads (causing a minor decrease in species-level OTU richness for the most diverse samples), and the full rarefaction curves of all samples. We note that even the most diverse samples exhibit saturation.

Referee #3:

R3.1. Dear editor, dear authors, I will perform a critical analysis point by point of each part of the manuscript with title "Airborne DNA metabarcoding reveals that fungi follow predictable spatial and seasonal dynamics at the global scale". This manuscript presents a significant contribution to the field of mycology and global biodiversity studies by leveraging airborne DNA metabarcoding to explore the spatial and seasonal dynamics of fungal communities across various climatic zones. The study stands out for its global scale and the application of advanced molecular and bioinformatic techniques to address a gap in current understanding of fungal distributions. The findings that fungal diversity exhibits predictable patterns related to climate, latitude, and seasonality are compelling and advance our knowledge of ecological and evolutionary processes shaping microbial life on a global scale. The rigorous methodology, comprehensive analysis, and clear presentation of results support the study's conclusions and highlight its potential to inform future research in ecology, conservation, and climate change studies. Overall, the manuscript is well-positioned to make a valuable impact on its field, subject to any specific revisions that might further clarify and enhance its contributions. The title is both informative and precise, clearly conveying the scope and findings of the study.

RESPONSE: We are grateful for this highly positive assessment.

R3.2. I do not have major concerns about the manuscript, just some suggestions. While the global scale of the study is impressive, there appears to be uneven geographic distribution of sampling sites, with potential gaps in some biomes. I understand the limitations of space required by the journal but a brief but precise acknowledgment of the study's limitations and suggestions for future research directions can significantly enhance the contribution. This might include a sentence or two on the geographic coverage of samples, potential biases in DNA metabarcoding, or areas where further methodological development could yield deeper insights.

RESPONSE: We fully agree about these limitations. As the limitations in geographic coverage were also noted by reviewer #2, we have added the following sentences to the Introduction and to the Methods:

"Although the European temperate region is overrepresented in the data, the sampling locations also include arctic, temperate, and tropical areas from other regions (Fig. 1A)"

"The reason for including only a small number of site-specific variables in the analysis is that, while the study is global in scope, it includes only 47 sites. The data thus hold limited information to statistically disentangle the effects of many spatially-varying covariates. Instead, the main strength of the study lies in its high temporal replication, which allowed us to identify effects of the spatiotemporal covariates, such as seasonality."

Concerning the biases related to the sampling method, we have added the following to the Discussion:

"However, some functional groups were better represented than others, highlighting the importance of surveying different complementary substrates to gain a complete view of fungal diversity"

R3.3. About the methodology, I appreciate the manuscript's detailed explanation of the optimized method for monitoring fungal spores. However, given the known presence of airborne hyphae in environmental samples, a verification step using an optical microscope for some samples, alongside a comparison with a Hirst sampler, could enhance the robustness of the findings. If such verification is not feasible, I recommend moderating the claims regarding the specificity of the sampling method to fungal spores, to acknowledge the potential for capturing other fungal structures.

RESPONSE: We thank the reviewer for this remark. In the revised text, we have now clarified that the sampler is not specific to fungal spores:

“A recent methodological breakthrough for the survey of fungi consists of sampling fungal spores (and other airborne particles, which may include fungal structures such as hyphae and soredia) from the air, followed by DNA sequencing and sequence-based species identification (Abrego et al., 2018).”

We however still use “spores” in the other parts of the manuscript, simply to avoid repeating the long explanation.

R3.4. As minor concerns, ensure that taxonomic names, locations, and technical terms are consistently capitalized and italicized as per the relevant scientific convention. This includes genus and species names in italics and consistent use, for example see Table 1 and Figure 2A.

RESPONSE: We thank the reviewer for spotting this inconsistency and have now harmonized the use of italics throughout the manuscript.

R3.5. I am not native speaker, but I found no one typographical error or mistakes referring grammar and punctuation. Neither related with references. Figures and tables have good resolution and are high quality.

RESPONSE: We thank the reviewer for checking the manuscript for potential errors.

R3.6. In conclusion, this manuscript presents a well-conducted and valuable study that significantly contributes to our understanding of the ecology of airborne fungal signals. The authors have addressed a relevant question with rigorous methodology and clear presentation of their findings.

RESPONSE: We are highly appreciative of this positive evaluation.

R3.7. I am unable to download the R scripts, even with the provided token. It seems that some additional configuration is needed. Data are public but restricted to users with access.

RESPONSE: We apologize for that. Before submitting the manuscript, we tested with another account that the link (copied below) should give full access. Nonetheless, when we retried it, we encountered the same problem as noted by the reviewer. We have now fixed the issue so the link should provide full access.

https://zenodo.org/records/10896659?token=eyJhbGciOiJIUzUxMiJ9.eyJpZCI6ImEzMTNhODgyLTQ0M2Q0tNGVhNi1hMTgzLTRhZTUzOTNkMDdlMyIsImRhZGEiOnt9LCJyYW5kb20iOiIxNTI0NmM0ZWU4NmUyZTRiZGRiNTVkNzJkZGI0MGVhZiJ9.YMBbe1Tp1MNB3vgrx3STrAcwiTn8aWgVkwGKT8lt4ctcUrf6WN5QroueoBqsjpT8t4W4_DuGye2UHh7S37yvvA

Reviewer Reports on the First Revision:

Referee #1 (Remarks to the Author):

Dear authors,

Thank you for very detailed consideration of all my (Referee #1) previous suggestions. Personally, I have no further comments on the manuscript.

Referee #2 (Remarks to the Author):

I appreciate that the authors took the time to respond to my comments and revise the manuscript accordingly. I think the manuscript is now much improved.

Referee #3 (Remarks to the Author):

The manuscript successfully demonstrates the use of airborne DNA metabarcoding to analyze fungal spore distributions globally, revealing clear spatial and seasonal patterns across different climatic zones. This methodology has shown considerable precision in identifying and quantifying fungal diversity, highlighting significant ecological insights.

As I stated in my previous review, this research is notable for its innovative approach to studying fungal distributions on a global scale. The findings contribute original insights into fungal ecology, particularly the predictable patterns of spore distribution and their environmental determinants, enhancing our understanding of fungal biogeography.

I am satisfied with the clarifications and answers of the authors, I support the publication of the manuscript. After being able to review the code and methodology, I agree that the methods applied, particularly the optimized sampling technique for spore collection, are rigorous and well-suited to the study's objectives. The data is robust, supporting the conclusions drawn.

The statistical analysis is thorough, with appropriate techniques used to analyze complex biological data. The conclusions are supported by the data and analysis, providing credible insights into the patterns of fungal diversity across various environments.

Referee #3 (Remarks on code availability):

I was able to download the code and carefully check all the scripts. I find the code very well organized, efficient and well documented. I don't find any problem.